# A probabilistic assessment of the rapidity of PETM onset

Sandra Kirtland Turner[1], Pincelli M. Hull[2], Lee R. Kump[3] & Andy Ridgwell [1,4]

Knowledge of the onset duration of the Paleocene-Eocene Thermal Maximum—the largest known greenhouse-gas-driven global warming event of the Cenozoic—is central to drawing inferences for future climate change. Single-foraminifera measurements of the associated carbon isotope excursion from Maud Rise (South Atlantic Ocean) are controversial, as they seem to indicate geologically instantaneous carbon release and anomalously long ocean mixing. Here, we fundamentally reinterpret this record and extract the likely PETM onset duration. First, we employ an Earth system model to illustrate how the response of ocean circulation to warming does not support the interpretation of instantaneous carbon release. Instead, we use a novel sediment-mixing model to show how changes in the relative population sizes of calcareous plankton, combined with sediment mixing, can explain the observations. Furthermore, for any plausible PETM onset duration and sampling methodology, we place a probability on not sampling an intermediate, syn-excursion isotopic value. Assuming mixed-layer carbonate production continued at Maud Rise, we deduce the PETM onset was likely <5 kyr.

[1] Department of Earth Sciences, University of California, Riverside, Riverside, CA 92506, USA. [2] Department of Geology and Geophysics, Yale University, New Haven, CT 06520, USA. [3] Department of Geosciences, The Pennsylvania State University, University Park, PA 16802, USA. [4] School of Geographical Sciences, University of Bristol, Bristol BS8 1SS, UK. Correspondence and requests for materials should be addressed to S.K T. (email: sandrakt@ucr.edu)

During the Paleocene-Eocene Thermal Maximum (PETM, ~56 Ma), the rapid injection of isotopically depleted carbon to the atmosphere (and/or ocean) was imprinted in the geological record as a prominent negative carbon isotope excursion. Associated with this is evidence for a ~5 °C global temperature rise, ocean acidification, and a variety of global biotic changes in marine and terrestrial archives[1–4]. The PETM is thus widely recognized as the best known analog to date for future greenhouse-gas-driven global warming[5]. However, the timescale of the event is critical to the value of inferences that can be drawn regarding future climate change and ecosystem response[6]—particularly with respect to the duration of main carbon release (PETM onset), which we define as the interval between pre-PETM carbon isotope values and the recorded carbon isotope minimum.

Existing estimates for the duration of PETM onset range from near instantaneous[7–9] to tens of kyr (e.g., ref. [10]), with the lower-end estimates proving particularly contentious[11–14]. Difficulties arise because independent age control is rare for shallow marine and terrestrial records, while in the deep sea, relatively slow sedimentation combined with bioturbation and changes in preservation act to smear out the signal in bulk sediments[2, 15–17]. Two recent studies attempted to forgo the need for an age model and instead placed age limits on onset duration on the basis of synchronicity between indicators of climate and carbon input[14] and relative carbon isotopic excursion (CIE) magnitude in different reservoirs[13]. However, there was no overlap in these estimates with ref. [14] inferring >4 kyr and ref. [13] <3 kyr. The duration of PETM onset and thus rate of carbon release hence remains highly uncertain.

In theory, measuring the carbon and oxygen isotopic composition of individual foraminifera tests should enable deep-sea records, with their attendant age models, to be used to detect the primary environmental change—through individual-level preservation of pre-, post-, and onset isotopic signals[8, 17, 18]—because of the short foraminiferal lifespan. These single foraminiferal approaches differ from typical paleoceanographic studies that measure multiple individuals in a single sample, which because of bioturbation and hence vertical displacement of the individual particles, acts to smear out the resulting measured signal[16, 19] and risks missing isotopic excursions of very short-lived or sparsely recorded events. To address this, ref. [8] generated high-resolution, single-foraminifera stable isotope records using mixed layer and thermocline-dwelling planktonic foraminifera from Ocean Drilling Program (ODP) Site 690 at Maud Rise in the Atlantic sector of the Southern Ocean—the most well-studied, expanded, and complete deep-sea PETM section drilled to date[20]. However, these data (Fig. 1a), measured on individual specimens >250 μm in diameter, raised two major conundrums. First, mixed-layer dwellers showed no so-called 'intermediate' δ13C values at the excursion onset as was expected for an onset occurring over thousands to tens of thousands of years. Rather, pre- and post-excursion specimens were separated isotopically by at least ~2‰ without intermediates. Second, there was a large stratigraphic offset (8 cm) between excursion δ13C values recorded first in mixed-layer dwelling *Acarinina* species and subsequently in thermocline-dwelling *Subbotina* species.

Interpreted at face value, the lack intermediate excursion values in the fossil record[8] suggests an extremely rapid (effectively instantaneous) onset to the PETM δ13C excursion and warming with a dramatic impact on global ocean circulation. In this view, there is a top–down transmission of the isotopic anomaly that is consistent with the hypothesis that atmospheric carbon injection triggered PETM warming. Based on the various age models for Site 690[21–24], the offset between surface and thermocline dwellers

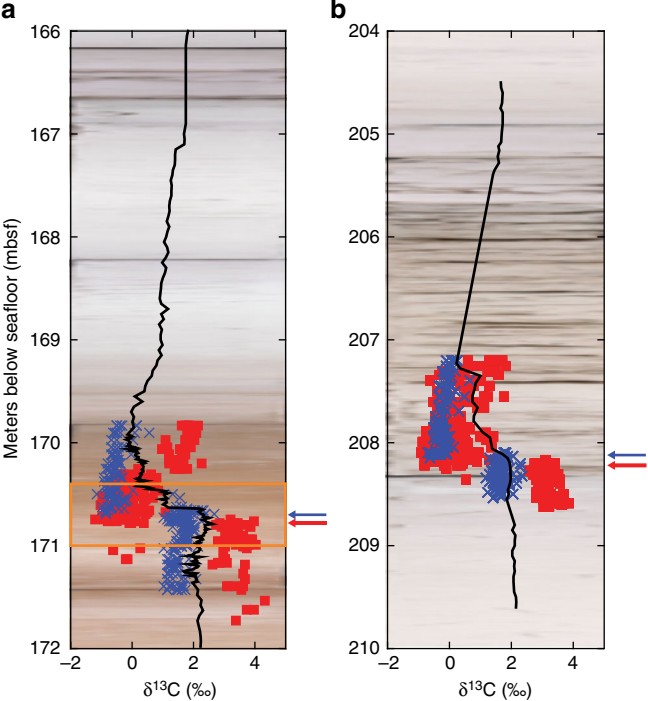

**Fig. 1** δ13C records from ODP Site 690 and ODP Site 689. Single foraminifer records for mixed-layer (*red*) and thermocline (*blue*) foraminifera across the PETM at **a** ODP Site 690 (ref. [8]) and **b** ODP Site 689 (ref. [17]). *Black lines* are bulk carbonate δ13C records from ref. [32]. Data are overlain on ODP core photos. *Red* and *blue arrows* indicate the onset of the δ13C excursion in mixed layer and thermocline dwellers indicated by ref. [17] for each site. *Orange box* in **a** highlights the interval expanded in Fig. 4a

translates to an apparent 5–15 kyr delay in the propagation of the perturbation from surface to thermocline waters[17]. This in turn implies extreme changes occurred in ocean vertical mixing in response to rapid carbon injection and warming across the PETM.

In alternative previous interpretations[17], sedimentary diagenesis and/or physical reworking, leading to a pronounced distortion of the signal encoded into the geological record, has been posited. Diagenetic over-printing has been discounted as a cause of this pattern because single-foraminifera measurements from nearby Site 689 (also located on Maud Rise but 1000 m shallower) have the same δ13C patterns between mixed layer, thermocline, and benthic foraminifera (Fig. 1b)[8, 17]. Preferential dissolution of mixed layer taxa is also considered unlikely to account for the full pattern[17]. This is because preferential dissolution would cause pre-PETM values in dissolution-resistant thermocline and benthic taxa to continue up section after the disappearance of pre-PETM values in mixed layer taxa (as is observed), but it should also result in the appearance of the first mixed layer excursion values only after those in the more dissolution resistant taxa[17]—the opposite of what is observed. Extensive sedimentary reworking has also been considered[25], given the evidence for bioturbation and possible winnowing in the core[26–28]. However, this is generally discounted as the sole explanation, largely on the basis of the preservation of abrupt step-like isotope shifts in the single species record[17, 20, 26]. However, we note that in addition to the similarity of the record at Site 689 (Fig. 1b) to the proximal 690 record, multiple other sites show the lack of intermediate values in single specimen mixed layer planktonic foraminiferal δ13C, including Pacific ODP

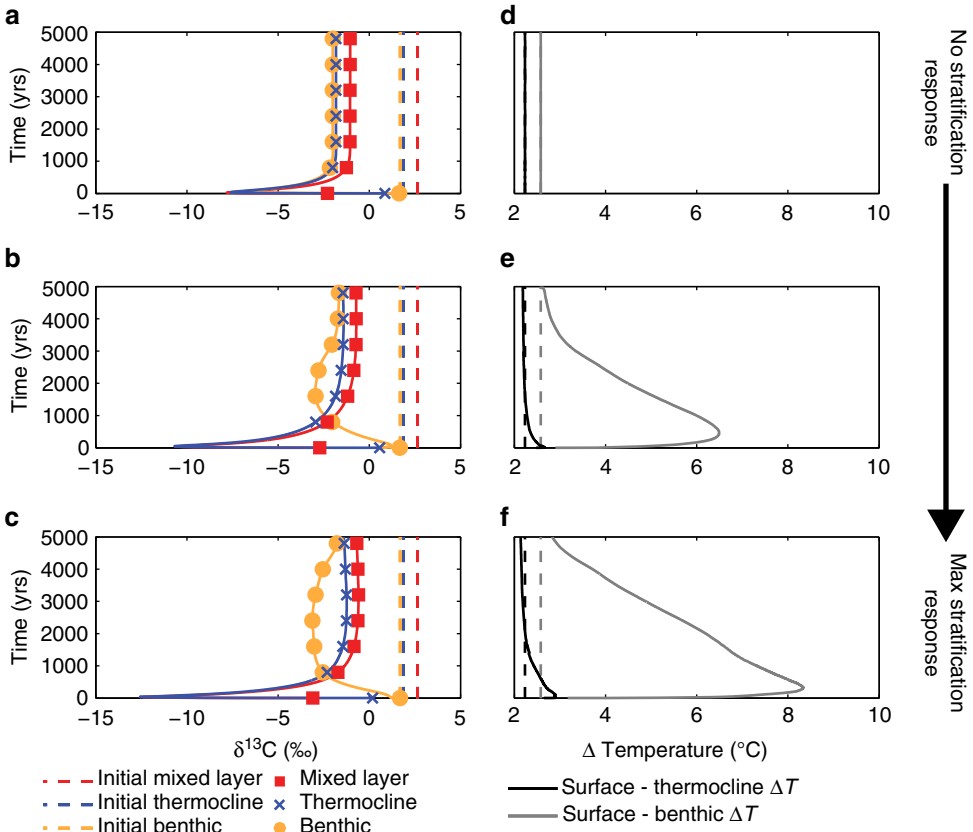

**Fig. 2** Enhanced stratification cannot explain substantial delays in propagating an instantaneous PETM $\delta^{13}$C excursion from the surface to the thermocline at ODP Site 690. **a**, **b** and **c** $\delta^{13}$C values recorded in the cGENIE mixed layer (*red*), thermocline (*blue*), and benthic (*orange*) ocean layers in response to injection over a single year of 2275 Pg C with $\delta^{13}$C of −60‰ to the atmosphere. This mass of carbon is sufficient to drive a −4‰ global $\delta^{13}$C excursion based on isotopic mass balance in cGENIE. Model location is equivalent to ODP Site 690 in the South Atlantic. Modeled thermocline and benthic depths are 128 m and 3283 m for Site 690. Fixed radiative forcing in each experiment controls the stratification response: at the equivalent of ×3 (i.e., no stratification response) **a**, **d**, ×10 **b**, **e** and ×25 **c**, **f** preindustrial $p$CO$_2$. *Symbols* indicate the record down-sampled at 800-year resolution, consistent with Site 690 data in ref. [8]. **d**, **e** and **f** Change in temperature difference between the surface and thermocline (*black*) and surface and benthic (*gray*). *Dashed lines* indicate initial temperature differences and solid lines indicate how temperature gradients evolve with time in response to the imposed temperature change

Site 865[18], North Pacific ODP Site 1209[29], and a continental shelf section in the Atlantic[17]. It is highly unlikely that disparate sites in both the Pacific and Atlantic basins and at both shallow and deep depths could be subject to the same diagenetic or sedimentary reworking artifacts.

Here, we show that differential changes in mixed layer and thermocline foraminiferal abundance across the PETM, combined with mixing of individual foraminifera (both down and up section) and limited sampling, can, in fact, account for both perplexing patterns at ODP Site 690 (i.e., lack of intermediates and diachroneity amongst groups). But we start by employing an Earth system model to test (and reject) the null hypothesis of instantaneous carbon release in conjunction with extreme ocean stratification as a viable explanation.

## Results

**Modeled ocean circulation and water column isotope response.** Using a late Paleocene configuration of the Earth system model cGENIE[30], we explored whether it was possible (at least in the context of the numerical model used) to induce a change in ocean circulation sufficient to create a sharp lagged response in the propagation of a $\delta^{13}$C anomaly though the ocean. We chose to employ an extreme scenario and apply a pulse of carbon released over a single year and sufficient to drive a −4‰ global $\delta^{13}$C excursion, following ref. [13]—not because we consider such a scenario likely (or even plausible), but to create a step change in

$\delta^{13}$C most reminiscent of the face-value interpretation of the data (Fig. 2). We also imposed varying degrees of surface ocean stratification in these experiments by adjusting model radiative forcing independently of the modeled CO$_2$ increase (see Methods section). The goal of these deliberately abstracted scenarios is to isolate the effect of enhanced warming while maintaining the same carbon cycle perturbation and hence fully elucidate the role of geochemical (carbon cycle) vs. physical (ocean circulation) changes. Because carbon release at the PETM was not necessarily directly to the atmosphere and *a priori* unintuitive pathways of tracer transport of an ocean release might be important, we also tested carbon injection directly to the intermediate ocean around continental margins, as if the source was the destabilization of marine methane hydrates[31], additionally testing an intermediate source only in the South vs. North Atlantic (Supplementary Figs. 1 and 2).

In the model output from all carbon release experiments, we analyzed the propagation of the $\delta^{13}$C anomaly at two locations at contrasting ends of the modeled deep-water flow path in order to evaluate the time required to propagate the $\delta^{13}$C excursion from the surface to thermocline at sites characterized by different degrees of vertical mixing (Supplementary Fig. 3). (For reference: in the unperturbed simulation of a late Paleocene global carbon cycle and climate in cGENIE, deep waters form in the South Atlantic (adjacent to Site 690) which is indicated by higher model deep ocean $\delta^{13}$C, with the water column weakly stratified there

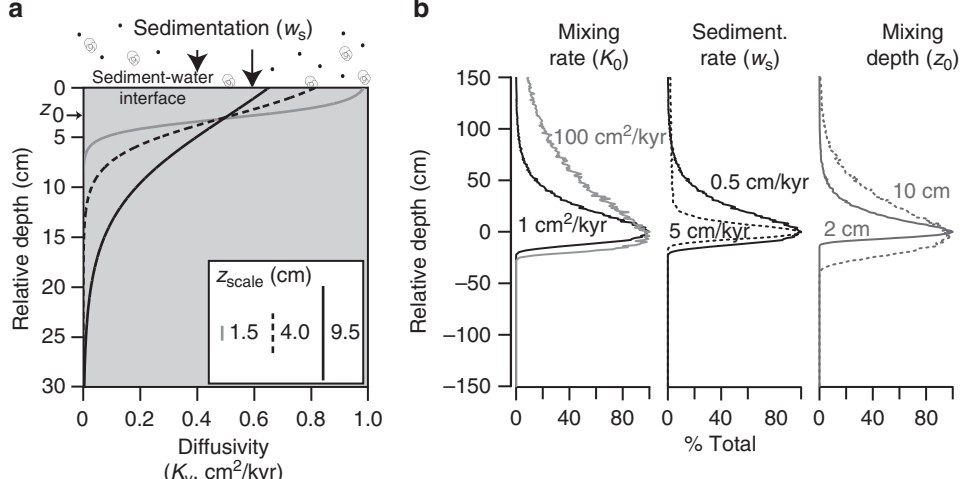

**Fig. 3** The Lagrangian sediment-mixing model and model behavior. **a** Lagrangian mixing model schematic after ref. [16]. The three *curves* show the effect of varying $z_{scale}$ on the depth- dependent diffusivity ($K_v$)–or mixing intensity. $z_0$ is held constant in these three scenarios and shown at the *left* with a *small arrow*. Other factors modifying the average mixing depth of particles is $K_O$ (maximum vertical diffusivity) and sedimentation rate ($w_s$). The effect of these factors on the average mixing profile of a point event is shown in **b**, a simplified version of a figure from ref. [16]

(Supplementary Fig. 3). In contrast, the North Atlantic, at the end of the deep-water flow path, which we include here in order to provide insights into possible behavior under a very different initial circulation state, is characterized by lower modeled deep ocean $\delta^{13}C$ and stronger vertical stratification (Supplementary Fig. 3)). We find that injection of carbon into the oceans rather than the atmosphere causes the excursion to appear first in either the thermocline or at the seafloor at model Site 690, inconsistent with the data (Supplementary Fig. 1) (see Methods section). For the North Atlantic site in the model, lying at the end of the deep-water flow path and where vertical stratification is greater, only carbon injection in the South Atlantic causes appearance of the excursion first in the surface, with significant delays in propagating the anomaly to both the thermocline and deep ocean (Supplementary Fig. 2, Supplementary Table 1). With carbon injection to the atmosphere, the arrival of the $\delta^{13}C$ excursion to depth is more delayed in the North Atlantic in comparison to modeled Site 690 near Antarctica. With increasing (imposed) radiative forcing, and thus enhanced vertical stratification (Fig. 2 and Supplementary Fig. 4), both locations show greater vertical propagation delays to the deep ocean. Yet crucially, under all tested degrees of radiative forcing (the end member of which leads to a global ocean surface warming of >8 °C in cGENIE, which is far in excess of observed warming across the PETM onset of 4–5 °C [1]), vertical stratification is insufficient to generate a delay in propagating the $\delta^{13}C$ anomaly from the surface to the thermocline of more than 20 years at model Site 690 or 300 years at model Site U1403 (Fig. 2, Supplementary Fig. 4, Supplementary Table 1, and Methods). Although no benthic single foraminiferal $\delta^{13}C$ record exists at this site, we note that significantly larger delays are possible before the excursion reaches the deep ocean (Supplementary Table 1).

Our Earth system model experiments thus show that it is possible to generate $\delta^{13}C$ inversions between mixed layer, thermocline, and seafloor depths, but these can only be sustained for decades to centuries between the mixed layer and thermocline (or a few thousand years between the deep and shallow sea) and not the 5–15 kyr interpreted at face value from the observations and existing age model.

**Modeled particle mixing and sedimentary isotope response**. Having discounted the viability of a 'face-value' interpretation of

the data implying a lagged top–down propagation of a near instantaneous atmospheric signal, and noting the unlikelihood that the primary features of the ODP Site 690 single-foraminifera isotope records reflect a diagenetic or sedimentary reworking artifact (see earlier discussion), we therefore turn to the dynamics of sedimentation and the non-intuitive consequences of bioturbation.

We employ a Lagrangian mixing model[16] to track the stratigraphic location of individual particles within a sediment column and simulate the formation of a complete sedimentary record of the PETM onset (see Fig. 3 and Methods section). In this, we start by treating the $\delta^{13}C$ excursion as an instantaneous step change in both 'surface' and 'thermocline' foraminifera with absolute $\delta^{13}C$ values for pre-PETM and excursion taken from Site 690 data. We assign an among-individual standard deviation to the $\delta^{13}C$ values of each population (surface or thermocline) of 0.30‰ and 0.12‰, respectively, based on variance in the ODP Site 690 data. With no other modification, both surface and thermocline records would look similar, with mixing carrying some foraminifera stratigraphically below their depositional depth, but predominately smearing individuals up-core into younger sediment (Supplementary Fig. 5). Given that this similarity between surface and thermocline isotope records at ODP Site 690 is not observed, we conclude that the abundance in the water column of thermocline vs. mixed-layer dwellers must have been differentially affected during the PETM onset.

We hence tested two approaches to modeling the combined effects of sediment mixing and differential abundance change on the single-foraminifera isotope record. (1) In the first approach, we made simplifying assumptions about the change in abundance of the thermocline-dwelling *Subbotina* and the mixed-layer dwelling *Acarinina* across the CIE onset. At ODP Site 690, carbonate content declined at the CIE onset from >85 wt% pre-PETM to lows of ~60%[32], indicating a marked reduction in the abundance of calcareous plankton in CIE sediments. Simultaneously, a large-scale faunal turnover occurred, with thermocline and mixed-layer dwelling taxa exhibiting pronounced fluctuations in relative abundance throughout the CIE[33]. To approximate these observed abundance changes, we modeled reductions in the abundance of mixed-layer dwelling taxa to half their initial population size at the PETM onset and thermocline-dwelling taxa to close to zero, with these changes occurring in a single step synchronous with the CIE onset (see Methods section, Fig. 4b–d).

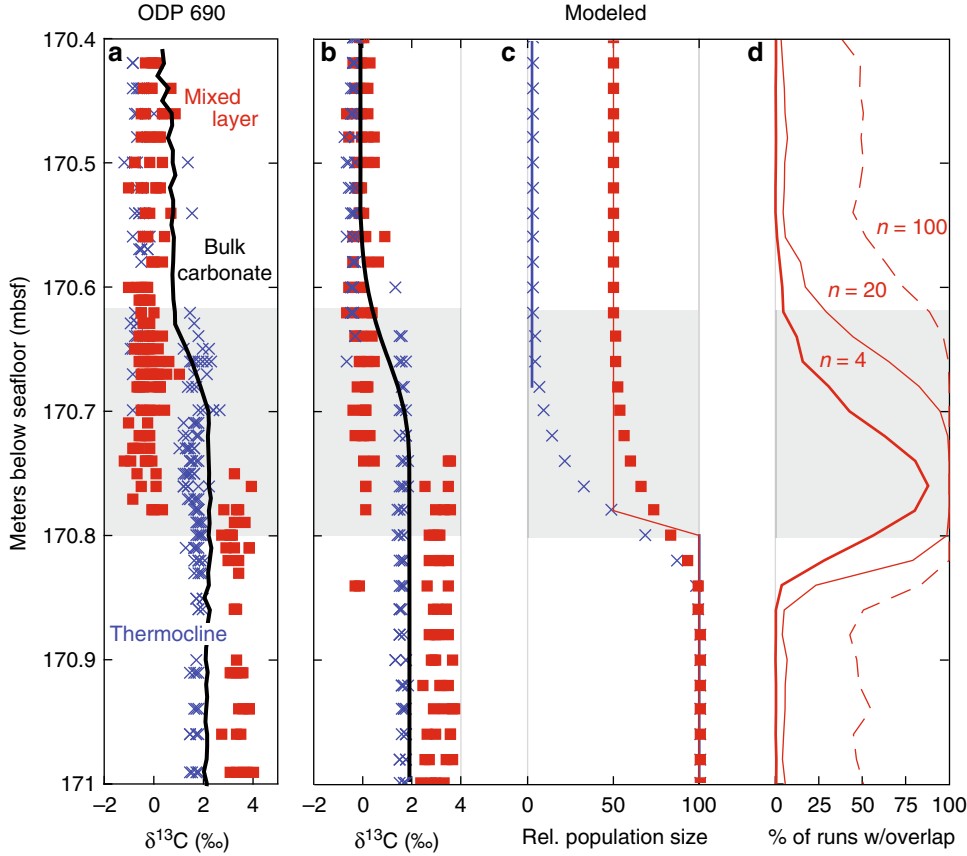

**Fig. 4** Sediment-mixing model including unequal changes in abundance across mixed-layer and thermocline species can explain patterns in ODP Site 690 data. **a** Single-foraminifera isotope data from ODP Site 690 across the PETM from mixed-layer planktonic foraminifera δ13C (*red*), thermocline δ13C (*blue*), and from bulk carbonate δ13C (*black*) (values from ref. [8]). **b** Sediment model simulation of δ13C in mixed-layer (*red*), thermocline (*blue*), and bulk carbonate (*black*) assuming a greater change in the abundance of thermocline species and in the nannoplankton taxa comprising the majority of the bulk carbonate than in mixed-layer species, with abundance changes indicated in panel **c** where *lines* = pre-mixing abundance and *symbols* = post-mixing abundance. The modeled onset of the CIE in the unmixed record occurs in all groups simultaneously at 170.8 mbsf—it is the combination of bioturbation and unequal abundance change that makes the δ13C change appear diachronous. **d** Despite bioturbation, most samples record just pre-event or CIE-type isotopic values when only small sample sizes (*n* = 4 individuals) are selected from the model record. *Gray box* highlights the interval in ODP Site 690 data showing the apparent delay in isotopic change between mixed layer and thermocline individuals

We used this model scenario to explore the general effect of sediment mixing on isotopic records in deep-sea sections with simplified abundance changes. (2) In the second approach, we used the best available data on the relative abundance change in *Subbotina* and *Acarinina* at ODP Site 690 across the target time interval[33], scaled for % foraminiferal sized fossils and % fragmented foraminifera in the sediment[32] (see Methods section, Supplementary Tables 2–4, and Supplementary Figs. 6 and 7), to generate a theoretical unmixed and mixed record of population abundance similar to the empirical data (Fig. 5b–e). We used this second model scenario to test whether the general hypotheses developed from the simplified model results (Fig. 4b) could account for the isotope records at ODP Site 690 given the observed population dynamics.

For direct comparison to isotopic observations (ref. [8]), we randomly selected four individuals from each of the two depth habitat groups every 2 cm from the theoretical sediment columns produced by the mixing model in both scenarios. Because of the stochastic nature of the model and random selection of four individuals every 2 cm, each model run results in slightly different single-foraminifera isotope patterns, of which Fig. 4b (and Fig. 5f) provide examples. Both the model simulations including realistic patterns of abundance change and the simplified mixing model scenario produce isotopic distributions like those observed

at ODP Site 690. Our modeling thus demonstrates that bioturbation, combined with a relatively greater reduction in abundance of thermocline taxa, can readily explain the delayed δ13C excursion in thermocline foraminifera. We can similarly reproduce the gradual decline observed in bulk carbonate isotopic values (Fig. 4a) with an abrupt reduction of nannoplankton populations synchronous with the PETM onset followed by bioturbation (see Fig. 4b and Methods section).

## Discussion

Because cGENIE incorporates a set of ocean physics[34, 35] that include more (e.g., frictional geostrophic) approximations than most conventional ocean general circulation models, there is the potential that stratification in response to rapid surface warming is being underestimated. In particular, cGENIE assumes a fixed vertical diffusivity in the ocean[34] meaning that diapycnal mixing is invariant to the degree of stratification and the rate of erosion of an induced density stratification will tend to be over-estimated. To test for an artifact in the ocean stratification response, we hence also repeated the Earth system model experiments using a stratification-dependent modification of vertical diffusivity[36] to enable a dynamic response of vertical mixing to abrupt global warming (see Methods section and Supplementary Figs. 8–10).

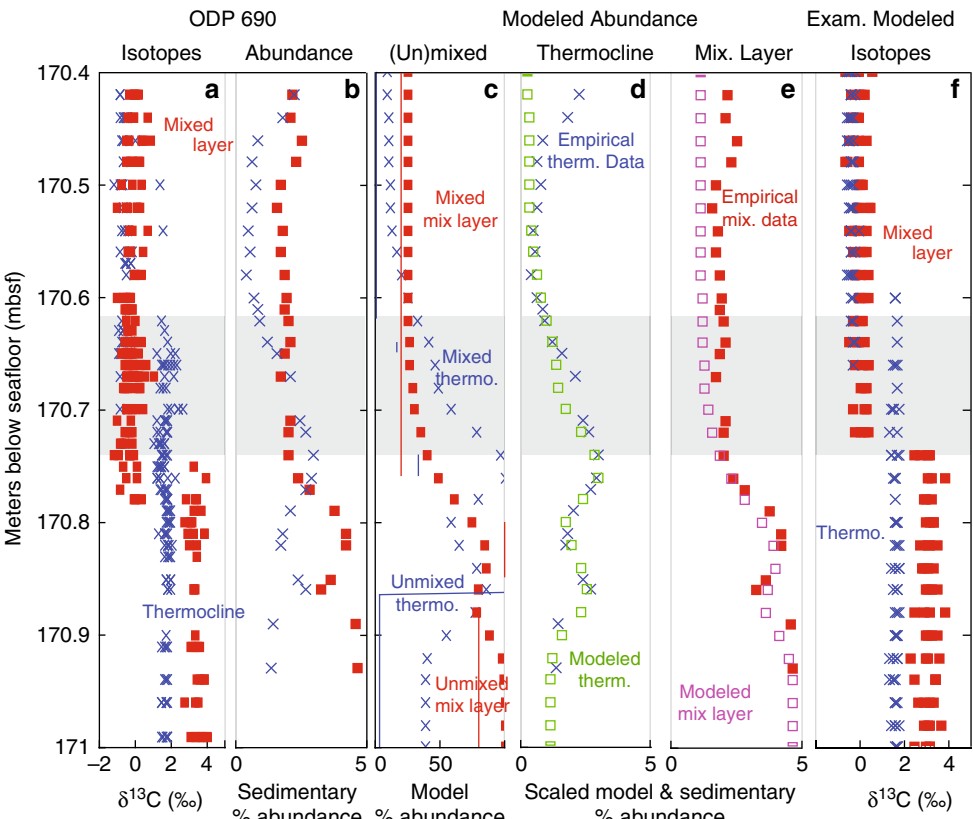

**Fig. 5** Sediment model with realistic abundance changes also generates isotope records matching patterns in ODP Site 690 data. **a** Single-foraminifera isotope data from ODP Site 690 across the PETM exhibit an apparently diachronous step change in mixed-layer planktonic foraminifera δ[13]C (*red*) and thermocline δ[13]C (*blue*) values from ref. [8]. **b** Sedimentary % abundance of the two relevant genera (thermocline: *Subbotina* (*blue*) and mixed layer: *Acarinina* (*red*)) varies throughout the interval and can be fit assuming numerous highs, lows, and gaps in the deposition of the two clades **c**. Mixed and unmixed populations of mixed-layer and thermocline genera were modeled initially with a maximum abundance of 100%, and then scaled to match the maximum abundance of the empirical thermocline data (**d**, *blue* = empirical data and *green* = model) and empirical mixed layer data (**e**, *red* = empirical data and *pink* = model). Once appropriate modeled abundances were obtained, CIE isotope onsets were applied at different depths to determine whether any depth of onset produced isotope records like those of ODP Site 690. An isotope change at 170.74 mbsf did **f**, with the *gray box* in each panel highlighting the interval in ODP Site 690 data with apparent delay in isotopic change between mixed layer (*red*) and thermocline (*blue*) individuals, with the lower edge of the box (at 170.74 mbsf) marking the modeled onset of the CIE in the unmixed records

We found that with mixing declining in response to differential surface warming, there was no significant impact on the surface-thermocline lag at Site 690, but slight increases in the surface to benthic lag (Supplementary Figs. 9 and 10, Supplementary Table 1). This modified representation of vertical mixing still cannot explain a long (5–15 kyr) lag in δ[13]C from surface to thermocline. Importantly, and in contrast to observations (Fig. 1), a δ[13]C step change at the surface caused by carbon injection to the atmosphere over a single year always transforms into a more gradual decline to minimum δ[13]C at depth, preserving intermediate δ[13]C values (Fig. 2, Supplementary Figs. 4, 9 and 10). We are therefore highly confident that changes in stratification and physical ocean mixing cannot account for the large stratigraphic delay in transmitting the δ[13]C excursion from the surface to the thermocline, even under the most extreme carbon release scenario and with amplified temperature change.

We did however find that significant delays in propagating the δ[13]C anomaly from the surface to thermocline (still less than ~2000 years) occur for a model location far from the source of overturning with significant vertical stratification and when carbon injection occurred directly into the ocean at a far removed location (Supplementary Fig. 2c and Supplementary Table 1). What is happening in this scenario is that CO₂ fairly rapidly

outgasses from the ocean surface close to the site of injection in the intermediate South Atlantic because the water column is relatively deeply mixed there. The δ[13]C anomaly induced in the atmosphere is then imprinted onto the surface ocean in the North Atlantic, but because of the local water column stratification, is only ineffectively propagated downwards. At the same time, an ocean δ[13]C anomaly propagates up the Atlantic at depth, along with the large-scale circulation in that basin, but much more slowly as compared to inter-hemispheric transfer in the atmosphere. Intermediate depths in the North Atlantic hence see a delayed δ[13]C minimum (via ocean circulation) as compared to the surface (via the atmosphere). However, while in this scenario the simulated propagation of a δ[13]C anomaly bears some resemblance to the data, we only find it occurring in a model location in the opposite hemisphere to where the observations originate. For this scenario to explain the data, the sense of net circulation in the Atlantic would have to be the opposite of that in the cGENIE model, which we discount on the basis of the consistency between projected and observed benthic δ[13]C gradients in the Atlantic (which reflect large-scale circulation)[15].

Previous studies may have underestimated the effect of sediment mixing as an explanation for the Site 690 and 689 single-foraminifera records because of the non-intuitive effects that mixing can have on the fossil record. For instance, the fact

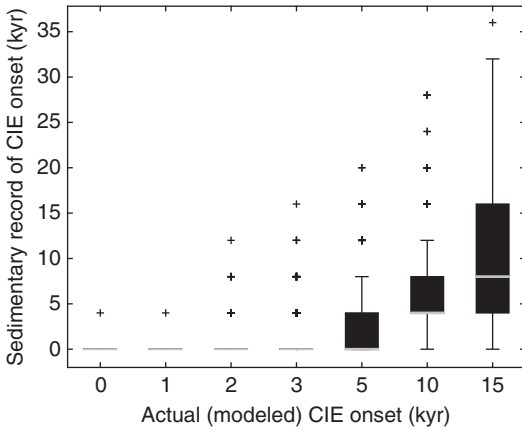

**Fig. 6** Sedimentary record can exhibit a step change from pre-event to peak CIE values in mixed layer individuals for CIE onset durations of 3 kyr or less. At a sampling intensity of 5-individuals per sample and a sample every 2-cm (sedimentation rates of 0.5 cm/kyr), *box and whisker plots* indicate the likelihood of the sedimentary record preserving an apparent 'instantaneous' onset, (i.e., lacking intermediate-value individuals). We assume that abundance of mixed layer species declines linearly from pre- to post-event levels over the entirety of the CIE onset (Supplementary Fig. 13) with no gaps in the deposition of mixed-layer taxa. For each *box plot*, the lower and upper edges correspond to the twenty-fifth and seventy-fifth percentile, respectively, with the median (fiftieth percentile) indicated as a *gray line*. Box plot whiskers span ± 2.7 s.d., with data lying beyond the whiskers shown as *black crosses*

that most samples from ODP Sites 690 and 689 contain just pre-event or post-event individuals has been taken as evidence for limited mixing (e.g., ref. [20] and ref. [26]). Yet despite a 10 cm well-mixed zone and the potential for re-entrainment to deposit a single sedimentary particle up to 25 cm above its initial depositional point in our model, we can generate records with only a modest number of overlapping individuals (Fig. 4b, d). Our small sampling size of just four individuals every 2 cm—while in fact slightly denser sampling than employed in generating the empirical records—results in near-exclusive sampling of the dominant sedimentary component. In order to sample two populations in a sample of just four individuals, populations must be roughly equally abundant.

Mixing provides an equally parsimonious explanation for the isotopic records even when the complex abundance histories of each clade are accounted for (as opposed to employing our highly simplified population change), as in Fig. 5. The primary difference between this scenario and the simplified model scenario (e.g., Fig. 4) is the inferred timing of the onset of the CIE in the unmixed records. In the simplified scenario, the CIE onset is modeled at 170.8 mbsf, a boundary coincident with a step change in the abundance of both populations (Fig. 4b) and isotopes. In the empirically fit population abundance scenario, we find that a CIE onset at 170.74 mbsf generates isotopic records (Fig. 5f) like those observed (Fig. 5a), given the population abundance histories of the two clades. The magnitude of population abundance change needed to fit the empirical abundance records across the early CIE were similar to those used in the simplified model—a roughly 80% reduction (as compared to 50%) in the population of *Acarinina* (Figs. 4b and 5c) and a 97.5% reduction (identical to the simplified model) in the population of *Subbotina* (Figs. 4b and 5c). More generally, we found that large-scale changes in relative abundance of these two species occurred throughout the record even well before the CIE onset (Fig. 5c). These large-scale changes in abundance were needed to fit the post-mixing highs and lows in observed

population abundance, and included numerous intervals of very low abundance (modeled as gaps for computational efficiency). Such dynamic changes in plankton populations are not unexpected, with dramatic range and abundance changes common in modern and ancient oceans[37–41]. Similarly, the differing magnitude of response in *Subbotina* and *Acarinina* to the CIE is consistent with previous observations of the dynamics of planktonic foraminiferal clades with differing ecologies and evolutionary histories. Different clades are known to exhibit divergent responses to the same environmental change throughout the Cenozoic[42], including the abrupt perturbations in the early Paleogene[43–46].

We emphasize that the observed record and mixing model track the abundance of unmixed and mixed populations in the sediment. As such, changes in the relative abundance of taxa reflect some combination of changes in living populations, body size distributions (i.e., size reductions remove individuals from the sampled size range), dissolution in the water column, and preservation in the sediment. In the case of ODP Site 690, many of these factors likely changed at the onset of the CIE as discussed in ref. [17]. Sediment mixing, well documented in most PETM sections, is also widely observed to change across the PETM—with a number of observations pointing to a reduction in bioturbation (e.g., ref. [26] and ref. [27]), and, in some sites, an increase in winnowing[26]. A reduction in bioturbation does not automatically imply a reduction in the average mixing range of a sedimentary particle. Instead, it is the interaction of bioturbation depth, intensity, and sedimentation rate[16]—all factors likely affected at the CIE onset—that determines the average mixing trajectory of a sedimentary particle (Fig. 3, Supplementary Figs. 11 and 12). With the low sedimentation rates due to reduced carbonate preservation occurring during the early part of the CIE, reduced bioturbation depths and/or intensities may have generated similar mixing trajectories to those in the pre-PETM (Supplementary Fig. 12).

The true post-depositional history of individual foraminifera and sediment grains is certainly more complex than any of the scenarios we have modeled here—including additional factors like size-dependent and/or lumpy mixing that we have not taken into account. However, what we clearly show here is that the dominant behavior of these isotopic records can be explained with a simple mixing model and a difference in the abundance change of taxa across the CIE onset. Thus while more complex sediment mixing scenarios may have occurred across the PETM in the South Atlantic, the explanation for the dominant patterns exhibited in the data, can, in fact, be easily simulated.

Assessing the likelihood of overlapping pre-event and CIE isotope values leads us to one final test—whether the lack of intermediate values in the Site 690/689 data can constrain the PETM onset duration. We used the simplest set of model assumptions, including constant sedimentation across the event and a single change in the abundance of mixed-layer species. We tested a ramped onset of the CIE over 0 kyr (geologically instantaneous), 1, 2, 3, 5, 10, and 15 kyr and for each (Supplementary Fig. 13), assessed the probability of detecting intermediate values given modeled sample sizes of 5, 10, 20, 50, 100, and 500 individuals in the simplified model (Fig. 6, Supplementary Fig. 14). We consider an intermediate value has been detected if we sample a value that falls outside 3-standard deviations of the pre-PETM and PETM values (all modeled samples were taken at 2 cm intervals). We then calculate the observed onset duration based on the number of consecutive samples with a least one intermediate-value individual in order to mimic the approach of ref. [8]. Our results show that onsets of 3 kyr or less are predicted to have no intermediate-value individuals in nearly all experimental runs, while 5 kyr modeled onsets record

intermediate individuals in just 50% of experiments (Fig. 6). Only for modeled onsets >5 kyr is the recovery of intermediate individuals predicted, given a sampling intensity comparable to the data (Fig. 6 vs. Supplementary Fig. 14). We conclude that while the lack of intermediate values recorded at Site 690 does not require an instantaneous PETM onset, the simplified model of the Site 690 record does suggest that the onset likely occurred over <5 kyr and not 10s of kyr. We thus also emphasize that the ecological response to global environmental change can itself impact how that change is recorded in proxies and hence interpreted. Future studies that seek to refine the estimate of the PETM onset duration should generate species-specific and size-specific counts of the same foraminifera targeted for individual specimen isotopic analysis at multiple deep-sea sites. The analysis presented here could then be repeated globally using differing model parameters appropriate for each setting. Overall, approaches such as this which combine data and new forward models until they converge (and see ref. [15]), are needed to progress forward in unambiguously interpreting past events and providing reliable insights into future climate change.

## Methods

**Mixing models.** To understand the effect of sediment mixing on the distribution of single-foraminifera isotopic values, we (i) modeled the effect of bioturbation on a single packet of sediment (that is, a point event) and then (ii) applied this average mixing profile to every 0.2 cm increment of sediment in the theoretical sedimentary column, assuming an original pre-mixed abundance and isotopic value for each 0.2 cm increment.

In step (i) we used the Lagrangian mixing model of ref. [16] with a sedimentation rate of 2.5 cm/kyr, a maximum vertical diffusivity ($K_0$) of 3 $cm^2$/kyr, a well-mixed layer ($z_0$) of 10 cm, and a mixing e-folding scale ($z_{scale}$) for the tanh profile of 1.5 cm, to mix a theoretical point event. These parameters matched typical sedimentation rates at the site in the late Paleocene to early Eocene and were informed by deep mixing of point event tracers at ODP Site 690 across the Cretaceous-Paleogene boundary[16]; other parameter combinations were explored (Supplementary Fig. 11) and some low mixing and low sedimentation rate cases generated similar mixing profiles to that described above (Supplementary Fig. 12). Other model parameters included a time step of 10 years, a mixing temporal duration of 60,000 years, a depth increment of 0.2 cm, and the inclusion of 10,000 mixing tracer particles. The full model was run 100 times in order to confidently calculate the typical (that is, median) mixing profile for a point event given this model and parameterization. The median mixing profile was then used in all subsequent steps. Although mixing rate changes likely occurred across the PETM onset[28], we aimed to model the mixing effective during the earliest phase of the CIE. This choice was practical as modeling mixing rate changes are computationally highly intensive in the modeling framework used and, without some constraint on the timing and extent of mixing change, would add a further source of un-parameterized model complexity. In addition, changes in mixing prior to our target interval (the CIE onset) will not affect the inferences made here about the effect of mixing on the geological record of the CIE onset.

To examine the importance of sediment mixing and changes in the relative abundance of thermocline and surface dwelling taxa, we made a series of simplifying assumptions in our mixing scenarios in step (ii). To start, we only model the profiles in single species $\delta^{13}C$ (and only at ODP Site 690), with the assumption that bioturbation would have a similar effect on $\delta^{18}O$ records. In the initial $\delta^{13}C$ simulations (Fig. 4), we also assumed a step change in the carbon isotope values of both species coincident with the onset of the CIE. 'Surface' foraminifera were assigned a mean pre-PETM carbon isotope value of 3.1‰ and a mean CIE carbon isotope value of 0‰, with an among-individual standard deviation of 0.30‰ throughout. 'Thermocline' foraminifera were assigned a mean pre-PETM carbon isotope value of 1.6‰ and a mean CIE carbon isotope value of −0.4‰, with an among-individual standard deviation of 0.12‰. In the simplified model, coincident with the CIE, the abundance of mixed-layer taxa was reduced by 50% and the abundance of thermocline taxa was reduced to near zero: 0 for the first 10 cm and 2.5% of the initial abundance thereafter. The 10 cm gap in thermocline taxa was used for computational efficiency: very low populations (less than 1%) were approximated by a gap in deposition. Modeled bulk carbonate (Fig. 4b) was modeled as the mean of a population experiencing an isotopic shift from 1.6 to −0.4‰ and a total abundance change to levels less than 1% of pre-event levels for 10 cm following the PETM onset and 2.5% of pre-event levels thereafter.

In the abundance fitting model, we derived empirical estimates of the change in the relevant thermocline and surface dwelling clades (i.e., *Subbotina* and *Acarinina*) from published counts, adjusted for the relative proportion of complete planktonic foraminifera in the sediment (Fig. 5). More specifically, relative % planktonic foraminifera species data of *Subbotina* and *Acarinina* from the >180 μm size fraction from ref. [33] were summed to get genus-level values. We used genus

level abundance data because multiple species were selected to generate the records in ref. [8] and varied over the duration of the record. Genus percentages were normalized to the total percent for each sample, due to reported values adding up to more than 100%. Normalized genus percentages were then multiplied by weight % coarse fraction and % foraminifera fragmentation data from ref. [32] to model the sedimentary % *Subbotina* and sedimentary % *Acarinina* (Fig. 5b, Supplementary Fig. 6). In many samples, weight % coarse fraction and/or % foraminifera fragmentation data were not available for the corresponding genus data. In these cases, we calculated the values using the relationship between weight % $CaCO_3$ and weight % coarse fraction (weight % coarse fraction = 0.1361*weight % $CaCO_3$−5.3177) and/or the relationship between weight % $CaCO_3$ and % foraminifera fragment (% foraminifera fragment = −0.3219×weight % $CaCO_3$+ 40.051) derived from the regressions of the relevant variables in the depth intervals of 170.31–171.42 mbsf (Supplementary Fig. 6). In the few instances in which weight % $CaCO_3$ data were also missing, the average of the adjacent depth intervals was used to calculate the parameter of interest. Because of the uncertainty introduced by the need to average and interpolate, a trend line was fit by eye to the weight % unfragmented foraminiferal data (i.e., weight % coarse fraction * % fragmented) (Supplementary Fig. 6) and this trend line was used to calculate the weight % unfragmented foraminifera for each sample with % *Subbotina* and % *Acarinina* data (e.g., Supplementary Fig. 7). Because of the numerous transformations needed, supplemental tables are provided with the data used and/ or calculated in each of these steps (Supplementary Tables 2–4). Best-fit model estimates of unmixed population abundance were then generated by iterative fitting and adjustment of unmixed populations to give mixed population size estimates matching the empirical data (Fig. 5c–e). Once good fits were derived, the step change in pre-event to CIE isotope values was applied at different simulated core depths between 170.8 and 170.7 mbsf, to see if a good match could be obtained between the abundance model scenario and the empirical isotope data (Fig. 5f).

The effect of varying the duration of the CIE onset was tested for onset durations of 0, 1, 2, 3, 5, 10, and 15 kyr, tracing the $\delta^{13}C$ of just the mixed-layer taxa, and using the sedimentation rate of 0.5 cm/kyr typical in the earliest interval of the CIE and the simplified model abundance scenarios. As in the previous experiments, the mixed layer acarininids were simulated with pre-event $\delta^{13}C$ of 3.1‰, CIE $\delta^{13}C$ of 0‰, $\delta^{13}C$ standard deviation of 0.3‰ throughout, and an abundance change of 50% from pre-event to CIE. Onset duration was tested for just mixed layer taxa as the inferred gap in thermocline species effectively precludes the CIE onset duration in the thermocline from being recorded in the sedimentary record. CIE onset duration experiments relied on several simplifying assumptions. Namely, mean $\delta^{13}C$ and individual abundance is assumed to decline linearly over the entirety of the CIE onset (Supplementary Fig. 13), 'intermediates' were detected only if they fell 3 s.d. outside of pre-event and CIE isotopic values, and onset duration was determined based on the number of consecutive samples with at least one intermediate-value individual. If population abundance declined more precipitously in response to the CIE onset, then our current simulation could lead us to infer shorter onset durations than likely occurred.

**Earth system modeling.** We used the cGENIE Earth system model to evaluate plausible delays in the propagation of isotopic anomalies with depth in the water column. The model includes a 3D dynamic ocean model with a simplified energy and moisture balance atmosphere and thermodynamic sea ice model[34, 41] coupled to a biogeochemical model[35, 42]. All experiments used a Paleocene configuration[27, 30] and were spun up as a closed system for 25 kyr prior to all perturbation experiments.

In order to represent an instantaneous PETM, we first applied pulses of carbon uniformly to the atmosphere over a single year and sufficient to generate a −4‰ $\delta^{13}C$ excursion in the cGENIE exogenic carbon reservoir (atmosphere plus ocean carbon)[13] using a light carbon source (−60‰, 2275 Pg C) to represent methane. Because model time series saving averages over an entire year, the first recorded values of surface ocean DIC $\delta^{13}C$ following carbon pulses are already lower than initial values from the spun-up model (indicated by dashed lines in Fig. 2, and Supplementary Figs. 1, 2, 4, 9 and 10). In order to quantify the influence of temperature-induced changes in ocean circulation on the propagation timescale of the isotopic anomaly, we further eliminated the modeled feedback between $CO_2$ and temperature and instead fixed radiative forcing at different values, first corresponding to no temperature increase (i.e., fixed at ×3 preindustrial $CO_2$, or 834 ppm), and then scaling radiative forcing as equivalent to ×10 and ×25 preindustrial $CO_2$ (Fig. 2). Although radiative forcing in these experiments is separate from the $CO_2$ increase, we note that it is crucial to simulate some increase in $pCO_2$ in order to correctly approximate the influence of air-sea gas exchange on the rapidity of the excursion onset in the surface ocean.

Next, we tested the impact of injecting carbon directly into the ocean. We applied pulses of 2275 Pg C with a $\delta^{13}C$ of −60‰ as dissolved inorganic carbon directly into the oceans in three different locations: either uniformly around the continents at intermediate depths of ~1000 m or localized at ~1000 m depth in just the North or South Atlantic Ocean.

For each experiment, we record the propagation of the isotopic anomaly through the 16 vertical ocean layers in the cGENIE model. The cGENIE ocean has a maximum depth of 5 km and vertical spacing that increases exponentially with depth from 81 to 765 m. We focus on two model locations: one corresponding to

the paleo-latitude and longitude of ODP Site 690 (65.7°S, 7.2° W) and the other corresponding to the paleo-latitude and longitude of IODP Site U1403 (39°56.60′ N, 51°48.20′W). These two model sites sit at opposite ends of the modeled deep-water flow path and are characterized by very different degrees of vertical stratification (Supplementary Fig. 3). We extract model time series for the carbon isotopic composition of dissolved inorganic carbon (DIC) and temperature for each model ocean depth at these locations. Surface values are those from the uppermost-modeled ocean layer and the thermocline is identified by the modeled vertical temperature profile at these locations (Supplementary Fig. 3), which shows a maximum d$T$/d$z$ at the ocean layer with a midpoint of 128 m for ODP Site 690 and at 346 m for Site U1403. For benthic values, we use the deepest ocean layer at each location, which has a midpoint of 3283 m for Site 690 and 4604 m for Site U1403.

For each experiment, we also employ the stratification-dependent mixing scheme of ref. [36] which substitutes a globally invariant value for diapycnal diffusivity ($2.5 \times 10^{-5}$ m$^2$ s$^{-1}$) with a modification of diapycnal diffusivity as a function of the deviation of the local density gradient from a reference density gradient profile obtained from globally averaged Levitus data[36]. Effectively, if the local density gradient exceeds this reference density gradient at a given depth, then diapycnal diffusivity is reduced relative to the default value ($2.5 \times 10^{-5}$ m$^2$ s$^{-1}$). Supplementary Fig. 8 demonstrates how diapycnal diffusivity at model sites 690 and U1403 corresponds to the default value when stratification-dependent mixing is employed. The more weakly stratified Site 690 shows significantly higher diapycnal diffusivity in comparison to the more stratified Site U1403. Both sites show decreases in diapycnal diffusivity in response to enhanced radiative forcing (and thus increased stratification), with more extreme impacts at Site 690, closer to the site of modeled deep-water formation, which decreases in response to increased stratification. Despite these changes, influence on the propagation of the δ$^{13}$C anomaly and warming is minimal (Fig. 2 compared to Supplementary Fig. 9 and Supplementary Fig. 4 compared to Supplementary Fig. 10).

We identify propagation timescale from the surface to thermocline by noting the difference between the times at which the full δ$^{13}$C excursion is reached in the surface and the thermocline (Supplementary Table 1). This method provides a maximum estimate for the propagation delay, since we are defining the excursion only when the minimum δ$^{13}$C value has been reached. This is not to suggest that no evidence of the excursion occurs over a shorter timeframe; excursions appear as gradual declines (rather than instantaneous step changes) in ocean layers away from the site of carbon injection.

**Data availability**. This study did not generate any new data. The data set analyzed here is from ref. [8] and can be obtained by contacting the corresponding author of that study. Model code required to reproduce the experimental results presented here can be found at:

https://svn.ggy.bris.ac.uk/subversion/genie/tags/2017.NC.690

(svn revision 9953) with the username genie-user and password g3n1e-user. Example configuration files for running necessary cGENIE spin-ups and experiments can be found in genie-userconfigs (files starting with EXAMPLE. p0055c.KTetal2017). Mixing model code can be found in genie-matlab/ KTetal2017_PMHmixingmodel.

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

## Acknowledgements

This study was funded by a Heising-Simons Foundation award to S.K.T., L.R.K., and A.R., with additional support to A.R. via ERC 2013-CoG-617313, and NSF Award #1536604 to P.M.H. We thank C. Kelly and D. Penman for insightful conversations during the early days of this research and E. Thomas for conversations in the later days.

## Author contributions

All authors contributed to the study design and final text. S.K.T. and A.R. developed cGENIE model experiments. S.K.T. conducted cGENIE experiments and analysis and P.M.H. conducted sediment-mixing model experiments and analysis.

## Additional information

**Competing interests:** The authors declare no competing financial interests.

