## [Peer Review File · Nature Communications]

Reviewers' comments:

Reviewer #1 (Remarks to the Author):

The Paleocene Eocene Thermal Maximum event at 55 million years before present is considered the best analog to modern global warming. For this reason the event has received an enormous amount of study from geologists over the last 25 years. Study has involved analysis of stable isotopic proxies, investigation of assemblages of planktonic calcifiers and application of coupled models. Much of the focus of study has been on deep-sea sites where investigations have uncovered the magnitude of the warming, associated changes in environment and the faunal and floral response. As the event involved input of massive amounts of carbon, it is associated with highly corrosive conditions in the ocean and dissolution of deep-sea carbonate. This results in condensed deep-sea sections, which prove to be difficult to study in the detail needed to extract millennial-scale signals. The PETM at a site on Maud Rise in the Southern Ocean, ODP Site 690 is less condensed than other deep-sea sites and thus has been studied in great detail. A paper that has received great interest published by Thomas et al. (2002) showed a sequence of shifts in single specimen foraminiferal stable isotopic values through the onset stage of the event progressing in time from foraminifera that inhabit the surface waters to those that reside at the thermocline. This top down sequence of events has also been observed in planktonic foraminiferal assemblages by Kelly (2002). The sequence is potentially important because it illustrates how the surface ocean warmed and how life responded. There is a major problem with the sequence however: available chronology indicates that it took 5-10 thousand years for warming to propagate from the surface down to depths of about 500 m, which does not agree with how heat is transported in the ocean: it is nearly impossible that the ocean could become that stratified. Moreover, there are no intermediate carbon isotopic values in the surface dwellers, which are expected if carbon input was slow. Relatively few single shells have been analyzed due to cost and this makes the results highly subject to mixing by processes at the seafloor. This dilemma has been puzzled over by numerous authors without satisfactory resolution.

The current manuscript describes a model-based investigation of the Site 690 record. The investigation has three separate components that are blended together in an elegant fashion: (1) application of the Earth System model to address the feasibility of stratification at Site 690; (2) application of a Lagrangian mixing model to the Site 690 single specimen record; and (3) a probabilistic analysis of the single specimen dataset to determine the duration of the onset of the CIE given the lack of intermediate C-isotope values. The manuscript is potentially a major breakthrough in our understanding of the PETM event as well as the introduction of a novel technique to study how single particles are mixed through bioturbation in any time interval. As such, I believe that the manuscript may justify publication in Nature Communications. However, I suggest that the authors need to think more about the key assumptions behind their mixing model as well as other processes at work at the seafloor at Site 690 during the PETM that would have impacted the single specimen record, how it is modeled and interpreted, and how the results are compared with independent datasets.

Overall, the manuscript is well written and generally logical for me to follow, although I indicate places below where the authors need to provide more explanation to readers who are unfamiliar with the Site 690 data as well as to those of us who are.

The manuscript begins with the application of an Earth system model to carbon input at the onset of the PETM. The authors show that even with input rates at the high end of current constraints for the PETM and maximum radiative forcing, the model is unable to sustain vertical stratification for the duration interpreted in the single specimen isotope record. This is a very fitting start for the manuscript as it sets up the sediment modeling perfectly.

A. The Lagrangian model and required population changes

The crux of the manuscript is in the Lagrangian modeling as this provides a potential explanation for the puzzling single specimen dataset. The dataset is highly complicated and it is very difficult to do justice to it in such a short manuscript, but I am quite close to the topic and when I finished reading I was confused as to exactly which specimens are thought to be mixed (this term used to combine bioturbated and winnowed) and which not, and more significantly, after their analysis, exactly where the authors propose to place the onset of the CIE in the surface dwelling single specimen record. Maybe the point is that it isn't possible to determine all of this exactly, but the onset level is the 64 million dollar question. The authors conclude that all thermocline specimens with pre-excursion C-isotope values above ~ 170.7 m are mixed upwards (lines 125-127) but do they conclude that all surface dwellers with excursion isotope values below this point (170.70-170.78 m) are bioturbated downwards or are they mostly in situ (I am guessing some of each)? The issue with placing the onset at 170.70 m is that the surface dwellers show no pre-excursion values between this point and ~ 170.76 m and this would require an explanation. This is ultra important as the shift in bulk carbonate (presumably nannoplankton) also begins at 170.70 m, and if the shift in surface dwellers (170.78 m) represents the true onset of the event, then why is the nannoplankton shift so late (note that the bulk shift is clearly impacted by bioturbation (see Bralower et al. 2014)? This would be counter to the results of the Earth system modeling. This exact point has confounded numerous previous workers (see Stoll 2005 for example) and must be cleared up here. It would help if samples that are proposed to be definitely out of place were differentiated in Figure 1 (it would also help if this figure had the bulk C-isotope curve). Moreover, the figure would be easier to follow with arrows indicating the proposed true onset of the CIE in the different fractions (panel a as proposed by Thomas et al., and panel b as proposed here).

I have four specific concerns regarding the model: (1) the nature of bioturbation at Site 690; (2) assumptions about starting foraminiferal species abundance (3) distinguishing between bioturbation and winnowing; (4) the impact of dissolution.

(1) The nature of bioturbation at Site 690 and the model parameters. The Lagrangian model has "a well mixed layer (z_0) of 10 cm, (line 170)" which needs to be compared with bioturbation in the record. The key core contains intervals with individual burrows generally less than 1 cm in diameter and shows changes in the intensity of bioturbation upwards. Some of the key interval is well mixed, some is not and some in between. Part of this is documented in an unpublished PhD thesis (Thomas, 2002) and in all fairness would not have been available to the authors, but the online core photos show some of the changes. Thus I recommend the authors devote some thought to how the model constraints match the nature of bioturbation in the core. Maybe I misunderstood the significance of z_0 but then this key parameter needs to be justified in more detail. How would the results be impacted if z_0 was 1 cm or 5 cm? What would happen if z_0 were to change at places in the core where sedimentary structures changed? Could changes in bioturbation have caused some of the single specimen disparities?

(2) Initial abundance of *Subbotina* spp. A critical assumption in the mixing modeling that is required to replicate the observed distribution of single specimen isotope values is that the abundance of the thermocline dwelling foraminifer *Subbotina* decreases to close to zero: "Ref. 29 inspected foraminifera in the >180 μm size range and found that thermocline-dwelling subbotinids were present but subordinate to acarininids during the bulk isotope CIE, with an earliest-CIE peak in subbotinas 29 likely mixed up from pre-event strata" (lines 110-113). Maybe I missed this, but I read Ref 29 a couple of times and could not find any such mention of mixing as an explanation for the abundance of *Subbotina*. Ref 29 does document a 50% decrease in the abundance of *Subbotina* at the onset of the bulk CIE but does not provide an explanation for a 95% reduction (reworking of specimens is discussed briefly in Zachos et al. 2007). Further the authors require a 10 cm hiatus in the deposition of thermocline specimens to make the model fit the data, which is also unsupported. Such reductions are not completely unfeasible, but the authors must provide a logical reason why *Subbotina* would be reduced so radically and then disappear. This genus appears to be fairly cosmopolitan and ubiquitous in the late Paleocene and early Eocene and one

way to explain its absence from an environmental viewpoint would involve a dramatic change in water column stratification whereby the thermocline was either extremely deep (via warming) or extremely shallow (via cooling). But this is counter to the results of the GENIE model. Regardless, the authors must justify this key assumption. Could the decrease in the species of *Subbotina*, *Subbotina patagonica* to 0 (Kelly 2002, Figure 4l) be significant? If so how? Even the results of random modeling of single specimens (Lines 127-129) that are "consistent with the observation that thermocline dwellers experience precipitous declines in abundance in the early CIE (e.g., Refs. 29, 30)" will not stand without a primary explanation. Alternatively, as I suggest below (3) and (4), the authors need to consider other more involved mechanisms of sorting. As written this is the weak link in the manuscript as the explanation of the single specimen data relies on it, and without a logical explanation for the decrease in *Subbotina*, this explanation will not stick.

(3) Distinguishing between bioturbation and winnowing. Bioturbation is an obvious process in the interpretation of single specimen isotope data. PETM foraminiferal shells weigh between about 5 and 20 micrograms and thus are subject to movement up and down in burrows. At the same time, the Site 690 sediments are likely to have been winnowed by bottom currents explaining their generally sandy nature (see grain size data in Bralower et al. (2014)). Although bioturbation and winnowing are both impacted by gravity, the Lagrangian model applied appears to be driven by bioturbation and I wonder if it can simulate both processes simultaneously. Winnowing involves resuspension of particles at the seafloor and thus may be a more effective means of size separation than bioturbation. This could be important as the combination of the two processes may help explain the stratigraphic sorting of *Subbotina* and robust *Acarinina* with bioturbation preferentially transporting heavier particles downward and winnowing preferentially mixing the lighter *Subbotina* specimens upwards. I suggest that the authors tweak their model to involve a skewed vertical mixing of particles to replicate the bias that winnowing might impose.

(4) The impact of dissolution. The authors assume a 10 cm hiatus in the deposition of thermocline specimens (Fig. 1c) (lines 118-119). As I mentioned in (2) above, I do not believe this is a realistic assumption without invoking a wholesale change in habitat. However, I wonder whether seafloor dissolution at the onset of the CIE masked by the heavy bioturbation could explain selective dissolution of the relatively fragile *Subbotina* relative to the robust *Acarinina*. It would be hard to imagine that all *Subbotina* specimens would be removed in this way, but I could envision a scenario whereby winnowing in concert with dissolution could be pretty effective in reducing the number of specimens radically.

In summary, the assumption that thermocline dwelling *Subbotina* was reduced to 2.5% of its initial abundance then disappeared for 10 cm is not realistic without invoking a dramatic change in thermal structure, which is exactly counter to the GENIE model. Because the outcome of the modeled data depends on this issue, the authors need to bolster this argument significantly. I recommend they consider additional mechanisms for the required reduction in *Subbotina* population including sorting via a combination of winnowing, bioturbation and dissolution.

B. Assemblage changes

This is a more minor issue and depends on the exact placement of the onset of the CIE in planktonics (see discussion above) whether at about 170.78 m or 170.70 m. Kelly (ref 29) documents a series of significant steps in planktonic foraminiferal assemblages, which suggest changes in habitat starting at the sea surface, progressing to the thermocline and ending at the seafloor. This is consistent with the Thomas et al. (2002) interpretation of single specimen isotope data. Moreover, given the large number of specimens counted in assemblage studies the trends are significantly more robust than those in single specimens, which as discussed here, are based on much rarer specimens. Obviously providing an alternative explanation for the assemblages is beyond the scope of this manuscript, but the authors should acknowledge that there is evidence for top to bottom changes in habitat, which requires an alternate explanation. If the true base of the PETM is at 170.70 m at the base of the bulk carbonate CIE, then several of the foraminiferal assemblage shifts occurred in the run up to the event and may not be directly associated with it.

This is a more minor concern than A, but it is a significant outcome of their interpretation and it needs to be mentioned.

In summary, I believe this manuscript will be a groundbreaking contribution. It has the potential to solve one of the most significant mysteries pertaining to the PETM while introducing a novel technique that can be used to unravel the complications associated with state-of-the-art single specimen isotopic analysis. I believe that once published the manuscript will instigate a new line of exciting investigation. It is not possible to cover every single base in a short communication. However, as mentioned above, there are currently several aspects of the analysis related to faunal population and sediment mixing that need to be strengthened significantly if the central hypothesis presented is to be convincing.

Minor Comments

Abstract: The second part of the abstract (lines 20-22) is written for someone who is familiar with the Site 690 dataset. Need to provide more information on the different foraminiferal isotopes.

Line 34 "constraint" should be plural.

Line 44. Here the authors need to provide a little more explanation why single foraminifera should be able to elucidate environmental changes and not multi-foraminiferal records.

Line 52. In fact the evidence for intermediate oxygen isotope values in surface dwelling foraminifera is pretty weak, only one specimen.

Line 67. This statement is not quite right. Grain size data suggest that sediments are winnowed.

Line 79. How was the modeled deep-water flow path determined?

Line 99 "We assign an among-individual standard deviation to the $\delta^{13}\text{C}$ values of each population (surface or thermocline) of 0.30‰ and 0.12‰, respectively." Do these values also come from Site 690 data?

Lines 122-129. I think this paragraph would be more powerful if there was a sentence stating that the pattern was similar to the observed data (Panel 1a).

Line 130-132. "Previous studies may have overlooked sediment mixing as an explanation for the Site 690 and 689 single-foraminifera records because of the non-intuitive effects that bioturbation can have on the fossil record." "Overlooked" is not an appropriate word here. The Site 690 PETM is clearly bioturbated and several studies have considered the impact of bioturbation on the single specimen record (e.g., Zachos et al. (2007) and Bralower et al (2014)). I would suggest, "underestimated the impact" as more appropriate.

Lines 377-78 Figure Caption for Figure 1. This diachronous step-change is readily simulated (b), assuming a greater change in the abundance of thermocline species (97.5%) than mixed-layer species (50%) (shown in panel (c)). It is hard to understand from this what these percentages mean.

Reviewer #2 (Remarks to the Author):

I very much would like to see a great paper emerge from this work, perhaps in Nature Communications. The authors have the talent and expertise to do this. However, I have mixed views on the current effort, and so I send a lengthy review of explanation for this perspective.

Sincerely,

Gerald (Jerry) Dickens

Prologue:

The PETM ca. 56 million years ago arguably represents our best past analog in which to understand rapid global warming and massive input of carbon to the ocean and atmosphere (Comment A). As might be expected for such an event, the causes and consequences remain the source of considerable interest and discussion across the broad Earth science community. Nature Communications should welcome good papers on the topic.

The current submission addresses broadly the timing of PETM carbon injection, and more narrowly, a puzzle regarding stable isotope analyses of individual foraminifera tests (shells) across the event. The PETM carbon input can be identified by a prominent negative carbon isotope excursion (CIE) in carbon-bearing phases. Individual foraminifera tests have been analyzed at several locations (including ODP Site 690 – the focus of the current study), but no tests have been identified so far with transitional $\delta^{13}\text{C}$ values (Zachos et al., Phil. Trans. Roy. Soc. A, 2007). The $\delta^{13}\text{C}$ of foraminifera are either pre-CIE or post-CIE (Comment B).

Some authors have interpreted the absence of transitional $\delta^{13}\text{C}$ values as signifying an extremely rapid input of ^{13}C -depleted carbon. However, as stressed by Zachos et al. (2007), there are at least three possibilities. Other than (i) “instantaneous” carbon input, there are (ii) dissolution of foraminifera tests because of corrosive waters on the seafloor (the original signal was removed), and (iii) diminished production of foraminifera tests because of environmental change in surface waters (an original signal was never generated).

The present manuscript sort of incorporates these thoughts into a numerical modeling perspective. The authors examine how a CIE should propagate through different reservoirs using a earth surface geochemical model (albeit with key assumptions, Comment A, below), and then, assuming that foraminifera tests faithfully record water chemistry through the onset of the PETM, determine how the CIE would be recorded using a sedimentation model.

I stress “sort of” because the manuscript neither fully sets-up the problem, nor fully discusses alternative possibilities (multiple comments below). A root question to ask: can a more complete and effective effort be placed into Nature Communications?

Basic Issues:

(A) Location and mode of carbon input.

(A1) It is by no means clear that the massive input of carbon CAUSED the pronounced warming and environmental change across the PETM. Multiple explanations for the carbon injection exist. Some of these, for example release of carbon from peat or seafloor methane, imply that the input was a feedback to warming.

(A2) It is by no means clear that the massive input of carbon ENTERED the atmosphere first. Some explanations, such as through dissociation of gas hydrate or North Atlantic sill intrusion, imply that much of carbon was added to the deep ocean.

The problem with the current effort is that plausible alternatives for carbon injection are not considered, and it is totally unclear how changes in basic assumptions would impact the modeling results and interpretations (Figures 2, 3).

On these matters, I note two items. There are transitional values in the $\delta^{18}\text{O}$ of planktonic foraminifera at Site 690, a point emphasized by Kelly et al. (Geology, 2002). These authors emphasized that this suggests warming led carbon injection, an idea also promulgated in several other works. Dickens (Bull. Soc. Geol. France, 2000), using a relatively simple geochemical model, demonstrated that responses of parameters ($\delta^{18}\text{C}$, carbonate dissolution) in different reservoirs, depends strongly on the direction of carbon propagation (e.g., deep-water flow) and the location of carbon input.

(B) Background to the manuscript.

(B1) The CIE needs better definition and explanation. This is because, in various papers, the CIE represents the entire excursion (onset through recovery), or alternatively, as in this manuscript, the initial drop in $\delta^{13}\text{C}$ (onset of the PETM). However, even the latter aspect is complicated, because multiple records suggest a complex decrease in $\delta^{13}\text{C}$ over time. Indeed, the latter point is a "root problem" largely omitted in the present manuscript. While en vogue to discount bulk carbonate $\delta^{13}\text{C}$ records, the fact that similar records occur at widespread locations, including some sites lacking bioturbation (e.g., Mead Stream, Nicolo et al., Paleocean., 2010), raises a provocative issue. Why are bulk carbonate records, such as at Site 690 and several other locations, showing a gradual, multi-stepped onset, when the foraminifera records exhibit no such change? (See Comment B2).

(B2) The "transitional foraminifera $\delta^{13}\text{C}$ puzzle" can be stressed better (see Zachos et al., 2007 and prologue above). More crucially, the alternative causes can and should be explored even further. What happens if the mixed-layer foraminifera examined for stable isotopes produced even less carbonate than modeled – in the end-member case – no carbonate during the onset of the PETM? Already, it is clear that multiple species of morozovellids and acarainids developed unusual morphologies near the start of the PETM.

The authors sort of note this point at the end of the Abstract (Lines 24-26), but then largely omit this crucial point from the rest of the manuscript. What if the mixed mixed-layer foraminifera examined for stable isotopes did not form carbonate during the onset of the PETM? This would solve many issues.

Other Problems:

(C) The modeling of carbon isotopes (Figure 2) needs better explanation in the text. The key point – and a good one – is that inversions in the mixed layer vs thermocline vs benthic $\delta^{13}\text{C}$ gradients can occur but cannot be maintained over time, despite stratification. However, I did not find this part crystal clear in the main text.

(D) The issue of bioturbation is not fully discussed. One of the authors (Ridgwell) has emphasized, and I think correctly, that one of the keys to understanding deep-sea records across the PETM (with most sites coming from "intermediate" paleo-water depths) is a reduction in bioturbation. Yet, it is not clear to me, if and how such a reduction was incorporated into the sediment portion of the modeling.

Specific Comments:

(I could probably make more on a revised manuscript)

Line 11: This sentence needs rewriting. There is zero evidence to state that greenhouse gases caused global warming during the PETM (Comment A1).

Line 14: Several foraminifera from the PETM record at Site 690 do have intermediate values in $\delta^{18}\text{O}$ but not in $\delta^{13}\text{C}$, as noted above. Adding a word "... associated carbon isotope ..." (Line 13) would help to clarify.

Line 24-26: The notion that mixed layer foraminifera carbonate production needs to continue across the PETM is a really important concept. It needs to be highlighted later in the text.

Line 29: Should be "led to a prominent negative carbon isotope excursion". This should be the very first point.

Lines 29-30: Following from above, there is zero support to state that [a] "... rapid injection of ... carbon ... LED to a $\sim 5^\circ\text{C}$ global temperature rise, etc." This is entirely wishful thinking with available data. "Associated with" is okay but "led" is not appropriate.

Lines 46-47: See above comment B1.

Line 50: I am not sure if this remains a correct statement. Maybe add "well-studied".

Line 65-67: Needs rewriting. There are multiple reasons for why this is not diagenetic. More importantly, the observation has been made at several sites.

Lines 75-79: This needs additional text. At the very least, it needs to be stated up front, that such modeling implicitly excludes certain possibilities for the PETM CIE excursion. (Ultimately, with a rapid carbon injection into the atmosphere from some "magical fossil-fuel source" that causes environmental change, they are invoking something absurd – like slow break-up of a giant methane-rich comet or a Mt. Pinatubo erupting every hour ...). Ideally, they fix and expand the framework to account for more reasonable possibilities.

Lines 85-86: It might be useful to consider the modeling by Dickens (2000), as a supporting conclusion might be derived from this work.

Line 92: Needs rewording. It is gradual in time but not magnitude.

Line 97-98: I do not follow this. Why should the thermocline change occur in conjunction with the mixed layer? I would think that, with the model assumptions on input, changes would necessarily lag those in the mixed layer, albeit by not very much time.

Lines 103-104: A sentence or concept is missing here to arrive at the conclusion. Something like "But this is not observed."

Line 109-110: I do not fully follow the phrase "mixed-layer dwelling foraminifera (referred to as 'robust' acarantinid variants in Ref. 29) and other acarantinids." There are two dominant genera of mixed layer foraminifera, morozovellids and acarantinids, and both have "excursion taxa" across the PETM.

Lines 110-113: I do not follow the point of this sentence, as it includes the word "bulk". This goes straight at comment B1. If the bulk carbon isotopes mean nothing, then how can be used to discuss stratigraphy?

Lines 122-123: The use of "sample" here is awkward. I assume a hypothetical sample.

Lines 137-138: This sentence on sample sizes, while important, is not crystal clear.

Line 148: As above, there needs to be clarification of "samples". In the current study, I think this

refers to "model space".

Lines 155-157: This conclusion is problematic and absolutely needs rewriting. This is because, as noted above, it depends (I think) entirely on model assumptions pertaining to (i) carbon input, (ii) incorporation of bioturbation, and (iii) the fidelity of the foraminifera $\delta^{13}\text{C}$ record.

Lines 159-161: I would suggest that such efforts also point out where such modeling needs serious challenge.

Lines 224-228: This is a fundamental problem with current cGENIE modeling. There is no means to add massive amounts of carbon rapidly without rubbing some bottle and invoking a magical source (perhaps why Andy called his model GENIE – ha ha), and no means to drive temperature without pushing from carbon. In particular, I do not think any one has suggested that $p\text{CO}_2$ increased to 10x let alone 25x pre-industrial values during the PETM. This should be stated as absurdly high. Basically, I can live with these problems in modeling papers for a few more years, as long as they are acknowledged. Of course, it would be better if they were fixed.

Figures:

In the figures, the benthic CIE appears magnified relative to the mixed layer. This is interesting (and probably important), because it is not observed in available $\delta^{13}\text{C}$ records.

For Figures S2 and S3: There is a time lag in temperature and $\delta^{13}\text{C}$ propagation through ocean. It should be stated that the ordering is because of model assumptions. (See Dickens, 2000).

Figure S4: I do not understand this figure conceptually. If all the carbon is added to the atmosphere to cause a 4 per mil excursion, then the excursion should be much, much less in the deep ocean once the carbon has propagated through the much larger ocean carbon reservoir (see for example, Zeebe et al. PNAS comment, 2014)..

Supplementary Information:

Line 6: This should be "... isotope records constructed using single foraminifera at ODP Site 690."

Reviewer #1 (Remarks to the Author):

The Paleocene Eocene Thermal Maximum event at 55 million years before present is considered the best analog to modern global warming. For this reason the event has received an enormous amount of study from geologists over the last 25 years. Study has involved analysis of stable isotopic proxies, investigation of assemblages of planktonic calcifiers and application of coupled models. Much of the focus of study has been on deep-sea sites where investigations have uncovered the magnitude of the warming, associated changes in environment and the faunal and floral response. As the event involved input of massive amounts of carbon, it is associated with highly corrosive conditions in the ocean and dissolution of deep-sea carbonate. This results in condensed deep-sea sections, which prove to be difficult to study in the detail needed to extract millennial-scale signals. The PETM at a site on Maud Rise in the Southern Ocean, ODP Site 690 is less condensed than other deep-sea sites and thus has been studied in great detail. A paper that has received great interest published by Thomas et al. (2002) showed a sequence of shifts in single specimen foraminiferal stable isotopic values through the onset stage of the event progressing in time from foraminifera that inhabit the surface waters to those that reside at the thermocline. This top down sequence of events has also been observed in planktonic foraminiferal assemblages by Kelly (2002). The sequence is potentially important because it illustrates how the surface ocean warmed and how life responded. There is a major problem with the sequence however: available chronology indicates that it took 5-10 thousand years for warming to propagate from the surface down to depths of about 500 m, which does not agree with how heat is transported in the ocean: it is nearly impossible that the ocean could become that stratified. Moreover, there are no intermediate carbon isotopic values in the surface dwellers, which are expected if carbon input was slow. Relatively few single shells have been analyzed due to cost and this makes the results highly subject to mixing by processes at the seafloor. This dilemma has been puzzled over by numerous authors without satisfactory resolution.

The current manuscript describes a model-based investigation of the Site 690 record. The investigation has three separate components that are blended together in an elegant fashion: (1) application of the Earth System model to address the feasibility of stratification at Site 690; (2) application of a Lagrangian mixing model to the Site 690 single specimen record; and (3) a probabilistic analysis of the single specimen dataset to determine the duration of the onset of the CIE given the lack of intermediate C-isotope values. The manuscript is potentially a major breakthrough in our understanding of the PETM event as well as the introduction of a novel technique to study how single particles are mixed through bioturbation in any time interval. As such, I believe that the manuscript may justify publication in Nature Communications. However, I suggest that the authors need to think more about the key assumptions behind their mixing model as well as other processes at work at the seafloor at Site 690 during the PETM that would have impacted the single specimen record, how it is modeled and interpreted, and how the results are compared with independent datasets.

Thank you. We found the points you raised very helpful in clarifying key aspects of the mixing model and results and in ensuring that the mechanism proposed (mixing) was consistent with existing data for species abundance and size distributions. We respond to the major and minor points in turn below.

Overall, the manuscript is well written and generally logical for me to follow, although I indicate places below where the authors need to provide more explanation to readers who are unfamiliar with the Site 690 data as well as to those of us who are.

The manuscript begins with the application of an Earth system model to carbon input at the onset of the PETM. The authors show that even with input rates at the high end of current constraints for the PETM and maximum radiative forcing, the model is unable to sustain vertical stratification for the duration interpreted in the single specimen isotope record. This is a very fitting start for the manuscript as it sets up the sediment modeling perfectly.

A. The Lagrangian model and required population changes

The crux of the manuscript is in the Lagrangian modeling as this provides a potential explanation for the puzzling single specimen dataset. The dataset is highly complicated and it is very difficult to do justice to it in such a short manuscript, but I am quite close to the topic and when I finished reading I was confused as to exactly which specimens are thought to be mixed (this term used to combine bioturbated and winnowed) and which not, and more significantly, after their analysis, exactly where the authors propose to place the onset of the CIE in the surface dwelling single specimen record. Maybe the point is that it isn't possible to determine all

of this exactly, but the onset level is the 64 million dollar question. The authors conclude that all thermocline specimens with pre-excursion C-isotope values above ~170.7 m are mixed upwards (lines 125-127) but do they conclude that all surface dwellers with excursion isotope values below this point (170.70-170.78 m) are bioturbated downwards or are they mostly in situ (I am guessing some of each)? The issue with placing the onset at 170.70 m is that the surface dwellers show no pre-excursion values between this point and ~170.76 m and this would require an explanation. This is ultra important as the shift in bulk carbonate (presumably nannoplankton) also begins at 170.70 m, and if the shift in surface dwellers (170.78 m) represents the true onset of the event, then why is the nannoplankton shift so late (note that the bulk shift is clearly impacted by bioturbation (see Bralower et al. 2014)? This would be counter to the results of the Earth system modeling. This exact point has confounded numerous previous workers (see Stoll 2005 for example) and must be cleared up here. It would help if samples that are proposed to be definitely out of place were differentiated in Figure 1 (it would also help if this figure had the bulk C-isotope curve). Moreover, the figure would be easier to follow with arrows indicating the proposed true onset of the CIE in the different fractions (panel a as proposed by Thomas et al., and panel b as proposed here).

We have made a number of changes to the text to hopefully obviate these questions for readers. We provide a response to each question here along with an indication of the changes made to the main text:

- ‘...I was confused as to exactly which specimens are thought to be mixed (this term used to combine bioturbated and winnowed)...’
 - We have not made an attempt to identify which individual foraminifera from the ODP 690 data are mixed. In the model, we apply mixing as affecting all sedimentary constituents. In our revised manuscript, mixing is applied to modeled mixed layer and thermocline foraminifera and to model bulk carbonate records. This mixing moves individuals both ‘up’ and ‘down’ in the sedimentary column, with the nature of the vertical displacement (up vs. down) and extent of displacement varying among sedimentary particles. Some particles move up and others move down, many effectively stay within the same cm interval, and a few are displaced by many tens of centimeters. Our previous description of the general nature of the mixing model was; ‘...bioturbation carrying some foraminifera stratigraphically below their depositional depth, but predominately smearing individuals up-core into younger sediment’ and in the Methods section ‘...we (i) modeled the effect of bioturbation on a single packet of sediment (that is, a point event)...’. In our revision, we have expanded the text regarding the mixing model in our results, discussion, and methods sections to clarify the modeling approach and findings.
- ‘...exactly where the authors propose to place the onset of the CIE in the surface dwelling single specimen record. Maybe the point is that it isn’t possible to determine all of this exactly, but the onset level is the 64 million dollar question.’
 - In model space, it is possible to place the onset of the CIE in the pre-mixed sedimentary column (i.e. where we model the CIE onset to generate panel (b) in Fig. 4 or panel (f) in Fig. 5). In our simplified model scenario (Fig. 4) the onset is modeled at 170.8 mcd (where the grey shaded bar begins) and in our model experiments where we have also attempted to fit abundance data (Fig. 5, again where the grey bar begins) the onset is modeled at 170.74 mcd. We now clearly state the modeled depth of onset in the figure captions and in the main text. In both sets of experiments, after mixing occurs (in the model), the modeled mixed layer, thermocline and bulk individuals can either be in place, above, or below their original stratigraphic position, but these displaced particles still carry the isotopic signature from their time of formation, as described in our discussion section and in the Methods. As we have emphasized in the text, our resulting mixing profiles are just one example of mixing model output. Therefore, while we can definitively identify the onset level that we apply in the model, it is not clear that we can precisely back out the onset level in the actual 690 data. Note that the model output (Fig. 4b and Fig. 5f) and data (Fig. 4a) are not an absolute simulation of reality; we can suggest that an onset of 170.8 mcd is consistent with mixing model results, but this is not the same as explicitly identifying the depth of PETM onset in the 690 record. This should now be clearer as an onset of 170.74 also produces model output consistent with observations when using a different patterns of abundance change.
- The authors conclude that all thermocline specimens with pre-excursion C-isotope values above ~170.7 m are mixed upwards (lines 125-127) but do they conclude that all surface dwellers with

excursion isotope values below this point (170.70-170.78 m) are bioturbated downwards or are they mostly in situ (I am guessing some of each)?

- As we described above, we have not specifically concluded which individual specimens in the Thomas record have been mixed and in which direction. Rather, both thermocline species and mixed layer species found in any given packet of sediment include individuals that are mixed up and individuals that are mixed down (and individuals that at the level of measurement, are effectively in place). In Figs. 4 and 5 we only model pre- and post- excursion values, so pre-event values at any depth were generated at pre-event model depths (pre-bioturbation depths of 170.8 mcd and deeper in Fig. 4 and 170.74 mcd and deeper in Fig. 5), regardless of where those particles ultimately ended up in the sedimentary column. From our model results in Figure 4b, the excursion 'starts' at 170.8, and the mixing up and down occurs across this interval (and across all species) as it occurs across all the other intervals modeled.
- The issue with placing the onset at 170.70 m is that the surface dwellers show no pre-excursion values between this point and ~170.76 m and this would require an explanation. This is ultra important as the shift in bulk carbonate (presumably nannoplankton) also begins at 170.70 m, and if the shift in surface dwellers (170.78 m) represents the true onset of the event, then why is the nannoplankton shift so late (note that the bulk shift is clearly impacted by bioturbation (see Bralower et al. 2014)?
 - We are not sure we entirely follow this comment. Our main conclusion in this manuscript is that the relative difference in abundance alters the apparent effect of mixing on isotopic records of different species and/or sedimentary components. We are not, in contrast, attempting to evaluate the precise placement of the onset in the 690 records or indicate which individual foraminiferal values have been displaced in which direction. We have reemphasized this point in our revised discussion. To address the question of how our mixing model impacts the bulk carbonate signal, we have added bulk carbonate (both data and model) to Figs. 1 and 4 and, using the simplified mixing model, show that the same model accounts of the shape of the bulk carbonate shift as well as key patterns in the foraminiferal data. The modeled onset (Fig 4b) is at 170.8 mcd before mixing, but the core CIE values dominate in the mixed records of different tracers (mixed layer, thermocline, bulk) at different depths due to the different abundance histories simulated for each tracer.
- This would be counter to the results of the Earth system modeling. This exact point has confounded numerous previous workers (see Stoll 2005 for example) and must be cleared up here.
 - We don't understand why our sediment mixing model results counter the Earth System modeling. Earth system modeling indicates near coincidence in the excursion onset between the surface and thermocline. The mixing model then provides a parsimonious explanation for the relative difference in timing of 'core CIE' isotope values as they appear in empirical records, even when the applied onset is equivalent for all tracers (mixed layer, thermocline, bulk). We have tried to make this key point clear in the abstract and throughout the discussion. We hope that this does indeed clear up some of the questions that have confounded previous workers, since this is the major aim of our manuscript!
- It would help if samples that are proposed to be definitely out of place were differentiated in Figure 1 (it would also help if this figure had the bulk C-isotope curve). Moreover, the figure would be easier to follow with arrows indicating the proposed true onset of the CIE in the different fractions (panel a as proposed by Thomas et al., and panel b as proposed here).
 - First, we have added a new figure (Fig. 1) that includes 690 and 689 data along with arrows indicating the identified onset from Thomas et al. in the mixed layer and thermocline dwellers. Next, the lower edge of the grey box in Figs. 4 and 5 indicates the modeled onset of the CIE in unmixed records (for all sedimentary components) as is now stated clearly in the captions and text. We want to emphasize that all samples may contain out of place individuals in both the empirical and modeled records. The majority of particles in any given packet of sediment in a real core (or modeled) are displaced to some extent (even if the displacement extent is below the resolution of the sampling). We thus cannot differentiate samples that are "out-of-place" because this would apply to all samples and most (but not all) individuals with isotopes measured (or modeled, as the case might be) in Figs. 4 & 5. It is important to recognize that the displacement of individuals is only noticeable where the isotope records are offset around the time of the CIE, but that just means that samples involved in this offset are the most noticeable examples of the displacement that impacts all samples. The 'out of place' nature of

many individuals is shared throughout the record – whether this is noticeable or not (it is only easy to see across the CIE). As such, the displacement of these samples is already noticeable, and we think it would be misleading to indicate them specially. As we note above, the true onset of the CIE is the same for all fractions in the unmixed records (bottom edge of grey box in Fig. 4 and Fig 5) and is now discussed. We hope the addition of the observed and modeled bulk carbonate record to Fig. 4 is helpful as well.

I have four specific concerns regarding the model: (1) the nature of bioturbation at Site 690; (2) assumptions about starting foraminiferal species abundance (3) distinguishing between bioturbation and winnowing; (4) the impact of dissolution.

(1) The nature of bioturbation at Site 690 and the model parameters. The Lagrangian model has “a well mixed layer (z_0) of 10 cm, (line 170)” which needs to be compared with bioturbation in the record. The key core contains intervals with individual burrows generally less than 1 cm in diameter and shows changes in the intensity of bioturbation upwards. Some of the key interval is well mixed, some is not and some in between. Part of this is documented in an unpublished PhD thesis (Thomas, 2002) and in all fairness would not have been available to the authors, but the online core photos show some of the changes. Thus I recommend the authors devote some thought to how the model constraints match the nature of bioturbation in the core. Maybe I misunderstood the significance of z_0 but then this key parameter needs to be justified in more detail. How would the results be impacted if z_0 was 1 cm or 5 cm? What would happen if z_0 were to change at places in the core where sedimentary structures changed? Could changes in bioturbation have caused some of the single specimen disparities?

- This was an excellent suggestion and we now include a paragraph expanding on our description of bioturbation in the 690 cores and in the model to complement the details already provided in Methods. We have added a new figure (Fig. 3) that summarizes the importance of model parameters and have also added further discussion about model assumptions. We show in our new model description figure (Fig. 3) that z_0 is just one parameter affecting the extent to which particles are mixed up and down core. It is possible to get the same average vertical displacement with very low z_0 (say, 3cm) if the mixing intensity is higher and/or the sedimentation rate lower (Supplementary Figure S12). To make the interaction of these three variables (mixing rate, sedimentation rate, and mixing depth) more intuitive in the context of this data set, we have added Supplementary Figures S11 and S12 showing different parameter combinations and the impact on the modeled 690 records. As we now discuss in the text and hope is clear from the new figures, the impact of changing z_0 on the extent of mixing depends on values assigned to the other mixing parameters. Without a point tracer of mixing in the 690 records (like an ash fall), it is impossible to parameterize the extent of mixing specifically. However, the parameter values used and average mixing profile generated in this study is reasonable for the Paleogene based on comparison with mixing models applied across the KPg boundary for similar deep sea sections, and is also reasonable given our understanding of modern mixing (now discussed directly in the Methods).

We also discuss what is known about mixing in PETM records generally – many records are well known to have a change in the nature of mixing around the PETM. In our case, the model assumes the mixing is constant across the CIE proper (see Methods) but a change in mixing from Pre-PETM to CIE will not affect our results (i.e., what matters is how much mixing occurs across the Pre-event/event boundary itself). We now try to make this point clear in the Methods. We apply the same model to all size fractions, so a change in mixing during the CIE wouldn't differentially affect the different size fractions (i.e., all sedimentary particles experience the same mixing) but could change the resulting mixing distributions and this in turn could change the apparent isotopic patterns between taxa given their different abundance histories. Because we can already fit the patterns with a simple model, we avoid adding multiple unparameterized factors including size-specific mixing, winnowing vs bioturbation-driven mixing, multiple changes in mixing intensity/depth and/or sedimentation rate, and lumpy mixing. While it would certainly be possible to fit models with higher complexity to the 690 data (more knobs to turn makes it easier to fit each and every data point) it is unclear what would be learned, as we wouldn't know whether those additional changes did in fact occur as modeled. What is clear is that a simple mixing model can fit the data. It is also clear that the true mixing history was almost certainly more complex than the history as modeled, but that a general class of 'mixing' explanations is sufficient to explain the salient features of the record.

(2) Initial abundance of *Subbotina* spp. A critical assumption in the mixing modeling that is required to replicate

the observed distribution of single specimen isotope values is that the abundance of the thermocline dwelling foraminifer *Subbotina* decreases to close to zero: “Ref. 29 inspected foraminifera in the >180 µm size range and found that thermocline-dwelling subbotinids were present but subordinate to acarininids during the bulk isotope CIE, with an earliest-CIE peak in subbotinas 29 likely mixed up from pre-event strata” (lines 110-113). Maybe I missed this, but I read Ref 29 a couple of times and could not find any such mention of mixing as an explanation for the abundance of *Subbotina*. Ref 29 does document a 50% decrease in the abundance of *Subbotina* at the onset of the bulk CIE but does not provide an explanation for a 95% reduction (reworking of specimens is discussed briefly in Zachos et al. 2007). Further the authors require a 10 cm hiatus in the deposition of thermocline specimens to make the model fit the data, which is also unsupported. Such reductions are not completely unfeasible, but the authors must provide a logical reason why *Subbotina* would be reduced so radically and then disappear. This genus appears to be fairly cosmopolitan and ubiquitous in the late Paleocene and early Eocene and one way to explain its absence from an environmental viewpoint would involve a dramatic change in water column stratification whereby the thermocline was either extremely deep (via warming) or extremely shallow (via cooling). But this is counter to the results of the GENIE model. Regardless, the authors must justify this key assumption. Could the decrease in the species of *Subbotina*, *Subbotina patagonica* to 0 (Kelly 2002, Figure 4I) be significant? If so how? Even the results of random modeling of single specimens (Lines 127-129) that are “consistent with the observation that thermocline dwellers experience precipitous declines in abundance in the early CIE (e.g., Refs. 29, 30)” will not stand without a primary explanation. Alternatively, as I suggest below (3) and (4), the authors need to consider other more involved mechanisms of sorting. As written this is the weak link in the manuscript as the explanation of the single specimen data relies on it, and without a logical explanation for the decrease in *Subbotina*, this explanation will not stick.

Our initial goal was to present the simplest possible model to show that mixing could explain the general isotopic patterns observed in the 690 records. Although we deliberately avoided the complexity of the data, we understand that the known history of species abundance changes can make our general approach and simplifying assumptions unsatisfactory if these assumptions appear to conflict with existing data. We have now directly addressed this concern with an additional mixing model scenario.

In this revision, we now include an example that fits modeled post-mixing abundances of mixed layer and thermocline species to match the empirical observations of *Subbotina* and *Acaranina* from ODP 690 (Fig 5). We explain how we derive the empirical observations of *Subbotina* and *Acaranina* from the literature in the text (and see Methods, Supp. Table S2, and captions to Supp. Figs. S6 & S7) and illustrate the relevant transformations in Supplementary Figures (Figure S6 & S7). We then show how the application of pre-event and peak-CIE isotopic values to these modeled ‘realistic’ abundance records still yields a mixed isotopic pattern that is highly similar to the empirical observations at 690 (Fig. 5).

This additional modeling exercise demonstrated several key points about the mixing model, including the approach and inferences, so we have devoted more space in the main text to discuss the results. These results include the observation that very large oscillations in taxa abundance (including multiple gaps) are needed to obtain records with relative abundance variations like that in the empirical records; thermocline taxa in both our simple model and our abundance fitted model have the same pattern of abundance change across the CIE onset including a decline in abundance of >95% and an immediate gap in deposition; the true abundance change that we apply to the model is (unsurprisingly) much greater than the observed post-mixing relative abundance change because mixing homogenizes the highs and lows.

In sum, we’ve now fit the *Subbotina* and *Acaranina* records of abundance and found an equally satisfactory mixing explanation for the isotope records, adding considerable robustness to our findings.

Because it is well known that different clades and/or closely related species with divergent ecologies often exhibit divergent responses to the same environmental event, we do not think a special, event-specific or top-down explanation is needed to justify the difference in relative abundance change between *Subbotina* and *Acaranina* at ODP 690 across the PETM onset. The two clades diverge ~10 million years before the PETM, have very different ecologies (*Subbotina*: inferred thermocline & asymbiotic; *Acaranina*: inferred mixed layer and symbiont-bearing), and tend to dominate assemblages in different locations prior to the PETM (e.g., *Morozovella* and *Acaranina* dominate in the Paleocene at Shatsky Rise, while *Subbotina* is relatively more abundant at Walvis Ridge). More importantly, there is a well documented rise of previously rare clades of symbiont bearing taxa in high latitude sites during the core CIE (e.g. Kelly 2002, see References) and this also suggests that some taxa and indeed clades may have found the warming at high latitudes relatively more

favorable than others. In addition, in fitting the pre-PETM variations in the abundance of *Subbotina* and *Acaranina*, we show that large oscillations in the relative abundance of taxa occur even before the event, adding support to the idea that peak event responses are expected to differ among taxa, even without invoking. By chance alone, abundance change should differ between clades. Across an event like the PETM, it is more likely that the response of these two clades would differ. By pure chance alone, the default expectation is that the population change between clades would differ randomly. We have now made these general arguments the primary explanation for the differing behavior of clades during the event. We therefore do not consider it necessary (or prudent) to evoke a specific mechanism for the difference in relative abundance change at the PETM given the context of the data considered in this study.

We discuss below the proposed addition of a separately parameterized winnowing effect but note here that winnowing is not needed to explain the abundance records of *Subbotina* and *Acaranina*. The generalized mixing model we employ (which sums effects like bioturbation and winnowing) is sufficient to explain the observed patterns.

(3) Distinguishing between bioturbation and winnowing. Bioturbation is an obvious process in the interpretation of single specimen isotope data. PETM foraminiferal shells weigh between about 5 and 20 micrograms and thus are subject to movement up and down in burrows. At the same time, the Site 690 sediments are likely to have been winnowed by bottom currents explaining their generally sandy nature (see grain size data in Bralower et al. (2014)). Although bioturbation and winnowing are both impacted by gravity, the Lagrangian model applied appears to be driven by bioturbation and I wonder if it can simulate both processes simultaneously. Winnowing involves resuspension of particles at the seafloor and thus may be a more effective means of size separation than bioturbation. This could be important as the combination of the two processes may help explain the stratigraphic sorting of *Subbotina* and robust *Acaranina* with bioturbation preferentially transporting heavier particles downward and winnowing preferentially mixing the lighter *Subbotina* specimens upwards. I suggest that the authors tweak their model to involve a skewed vertical mixing of particles to replicate the bias that winnowing might impose.

Both bioturbation and winnowing are known to be size selective processes and both can be modeled with advection diffusion type models like that utilized here. Including winnowing (as suggested) would simply require adding a second advection-diffusion model with size-dependent differences in the extent of transport. Because we can fit the records with a single advection diffusion model (Fig 4 and 5), we did not want to unnecessarily increase the complexity of the model. Instead, we simply reemphasize in the text that more complex mixing histories in all likelihood characterize the true history of the section, but that a simple size-unselective model is sufficient to explain the results. We also note that the effect of our simple mixing model is to predominately mix individual particles up-section (not downward; note the asymmetry in mixing profiles in Figure 3).

We now mention winnowing explicitly in the text to emphasize that this process has been suggested to affect various PETM sections.

By discussing alternative mixing scenarios and the mixing model more directly in our revision, it should be clear that our findings do disagree with the discussion in Bralower et al. EPSL 2014 (see References), although we do not state this directly in the text. In that paper, the authors argue that potential explanations of the offsets in isotopic records at 690 could be driven by 'bleaching', winnowing or bioturbation, with bioturbation alone being an unlikely explanation. Here we show that size-unselective mixing can in fact account for the observed isotopic patterns. Our results emphasize a well known feature of natural records: patterns that appear complex do not necessitate complex drivers.

(4) The impact of dissolution. The authors assume a 10 cm hiatus in the deposition of thermocline specimens (Fig. 1c) (lines 118-119). As I mentioned in (2) above, I do not believe this is a realistic assumption without invoking a wholesale change in habitat. However, I wonder whether seafloor dissolution at the onset of the CIE masked by the heavy bioturbation could explain selective dissolution of the relatively fragile *Subbotina* relative to the robust *Acaranina*. It would be hard to imagine that all *Subbotina* specimens would be removed in this way, but I could envision a scenario whereby winnowing in concert with dissolution could be pretty effective in reducing the number of specimens radically.

Subbotina and *Acaranina* populations can decline as a result of a primary change in the population size or in preservation potential. Our model is agnostic as to which applies -- we now emphasize this in the text. There are potentially a variety of environmental changes that could have differentially impacted *Subbotina* -- for

instance, a greater reduction in the water column saturation state with respect to calcite at the depth habitat of *Subbotina* while the surface ocean remained oversaturated. Such a change is plausible because the thermocline already has a lower saturation state with respect to calcite than the surface ocean due to organic matter remineralization in the water column. This remineralization may have intensified due to increasing temperatures across the PETM, leading to even less hospitable water column chemistry for thermocline dwellers. However, such specific scenarios to account for relative abundance changes are beyond the scope of our current study.

We disagree, in general, that it is unrealistic to evoke large-scale changes in population size or geographic range as a mechanism, given that this is very well documented in planktonic foraminifera across other events and across the PETM. As discussed above, we have added discussion in the text emphasizing that our findings of highly dynamic plankton population dynamics are perfectly fitting with the known dynamics in these groups.

In any case, we found directly modeling the *Subbotina* abundance changes to be very helpful and this exercise considerably strengthened the manuscript, so we are grateful for the suggestion. We have worked to emphasize the potential of winnowing and dissolution within the confines of our modeling and the empirical data.

In summary, the assumption that thermocline dwelling *Subbotina* was reduced to 2.5% of its initial abundance then disappeared for 10 cm is not realistic without invoking a dramatic change in thermal structure, which is exactly counter to the GENIE model. Because the outcome of the modeled data depends on this issue, the authors need to bolster this argument significantly. I recommend they consider additional mechanisms for the required reduction in *Subbotina* population including sorting via a combination of winnowing, bioturbation and dissolution.

We believe that our expanded modeling (including abundance changes) and explanations show that our proposed mechanisms are consistent with the data from ODP 690 on abundance changes in *Subbotina*. While we can hypothesize a number of reasons for differential abundance changes between different clades with very different ecologies (e.g. impact of saturation state outlined above), without additional Earth system modeling, these hypotheses would be highly speculative and therefore we do not believe that they would add substantively to our current effort. We choose instead to focus on the impact that abundance changes would have on the sedimentary record -- the main purpose of this study. We hope these points are now clear in the revised manuscript.

B. Assemblage changes

This is a more minor issue and depends on the exact placement of the onset of the CIE in planktonics (see discussion above) whether at about 170.78 m or 170.70 m. Kelly (ref 29) documents a series of significant steps in planktonic foraminiferal assemblages, which suggest changes in habitat starting at the sea surface, progressing to the thermocline and ending at the seafloor. This is consistent with the Thomas et al. (2002) interpretation of single specimen isotope data. Moreover, given the large number of specimens counted in assemblage studies the trends are significantly more robust than those in single specimens, which as discussed here, are based on much rarer specimens. Obviously providing an alternative explanation for the assemblages is beyond the scope of this manuscript, but the authors should acknowledge that there is evidence for top to bottom changes in habitat, which requires an alternate explanation. If the true base of the PETM is at 170.70 m at the base of the bulk carbonate CIE, then several of the foraminiferal assemblage shifts occurred in the run up to the event and may not be directly associated with it. This is a more minor concern than A, but it is a significant outcome of their interpretation and it needs to be mentioned.

We agree that providing an alternative explanation for assemblage changes is beyond the scope of this study. For that reason, we have not included a discussion of this topic in the revised manuscript. Here, we explain in detail.

Fig. 5 shows the results of our efforts to fit the population abundance swings documented by Kelly 2002 combined with Kelly 2012 (i.e., % abundance change modified to % complete foraminifera per unit of sediment, see References). We were able to make generic-level fits of abundance that were 'spot on' (Fig5d&e). With these fitted abundances, we then applied the pre- to peak-CIE isotope change at different depths to see if we could obtain isotope records like those from Thomas et al. 2002. The answer was 'yes' (Fig 5f) when we made the CIE onset start at 170.74 mcd (bottom of the grey box in Fig 5). This depth is stratigraphically above the

interval where the top-down changes in species abundance were recorded, so we do not attribute these dynamics to the CIE. We think it reasonable that abundance changes of various sorts occurred throughout the fossil record (indeed there is another distinct peak/trough at about 170.87 mcd), so we agree that several assemblage shifts occurred in the run up to the PETM that may not be directly associated with the event.

The reality is of course more complex. Currently this manuscript includes two very different abundance-change scenarios that both give the same (more-or-less) isotope pattern that match the ODP 690 data. Because the abundances changes differ, the inferred onset of the CIE in unmixed records differs. We would derive yet another unmixed population abundance history for *Subbotina* and *Acaranina* (e.g. Fig 5c, solid lines) if we reduced the mixing depth and intensity (Fig. S11). Our point is that there are an infinite number of alternative mixing scenarios, and for many of them, it would be still be possible to fit the observed abundance fluctuations (Fig. 5b). For at least some (if not all) of these well-fitted scenarios it would further be possible to produce reasonable isotope profiles (like those in Fig. 4b and Fig. 5f) and for those the depth of probable CIE onset might differ from the 170.74 mcd determined above. Without knowing the extent of mixing (which we cannot without a point tracer like an ash fall) it is hard to discern amongst these options for the exact onset of the unmixed CIE.

However, there is a very large source of uncertainty at present in the relative abundance data. As Thomas shows in her thesis, bulk isotope records generated from two halves of the same ODP 690 core differ...as might be expected from randomly sampling different parts of burrows in the different halves. How this type of uncertainty affects the foraminiferal counts is unquantified. Also unquantified is the relative change in % abundance encountered between foraminiferal counts on the same sample. Furthermore, the published abundance counts from Kelly, 2002 (see References), available on Pangea summed to more than 100% (by varying amounts), requiring normalization to the sum percent for each sample, but it is unclear why this problem exists. More problematic still is that the counts were carried out on the greater than 180 micron size fraction whereas the isotope data were generated from the greater than 250 micron size fraction. Matching abundance and isotopic records precisely in order to determine the exact depth of the CIE onset – and whether habitat changes did occur from the top down – would require recounting relative abundances at the greater than 250 micron fraction only. There is also the issue that fragmentation counts and coarse fraction data do not coincide with the species counts for many samples and had to be interpolated (see Methods). Also, the species of *Acaranina* and of *Subbotina* picked for isotopes varied across the record, but we don't have this list, so we can only fit the genus-level records. Again, we would need to simulate this clade-hopping in a more complex abundance fitting exercise in order to try to precisely back out the CIE onset.

While it is theoretically possible to get a more precise (with-error bars) estimate on the timing of the onset of the CIE in unmixed records and determine whether or not the habitat change was sequential from surface to deep, it is simply not possible to do so using the available published data. Unfortunately, because of the great importance of ODP Site 690 and the subsequent use and destruction of samples, it probably isn't possible to ever generate the necessary data. Rather than elaborate on these questions in the manuscript, we have simply explained here why we can't and why (likely) no one can answer these questions rigorously. Hopefully by showcasing the power and importance of the sediment mixing approach to the broader community, we will encourage collection of the necessary types of data (comparable abundance and isotopic records) using future cores.

In summary, I believe this manuscript will be a groundbreaking contribution. It has the potential to solve one of the most significant mysteries pertaining to the PETM while introducing a novel technique that can be used to unravel the complications associated with state-of-the art single specimen isotopic analysis. I believe that once published the manuscript will instigate a new line of exciting investigation. It is not possible to cover every single base in a short communication. However, as mentioned above, there are currently several aspects of the analysis related to faunal population and sediment mixing that need to be strengthened significantly if the central hypothesis presented is to be convincing.

Thank you again for this thoughtful and helpful review. We hope that the addition of new figures, Supplementary Table 2, and extra discussion in the text have fully addressed your concerns!

Minor Comments

Abstract: The second part of the abstract (lines 20-22) is written for someone who is familiar with the Site 690 dataset. Need to provide more information on the different foraminiferal isotopes.

We have modified the sentence to build on the information given previously in the abstract as

"We then use a novel sediment-mixing model that tracks individual particles to show how changes in the relative population sizes of the calcareous plankton used to track surface to deep ocean conditions, combined with sediment mixing, can explain the observations."

Line 34 "constraint" should be plural.

This sentence no longer exists as previously written, so the correction does not apply.

Line 44. Here the authors need to provide a little more explanation why single foraminifera should be able to elucidate environmental changes and not multi-foraminiferal records.

Added!

'These single foraminiferal approaches differ from typical paleoceanographic studies that measure multiple individuals in a single sample and thereby miss isotopic excursions of very short-lived or sparsely recorded events.'

Line 52. In fact the evidence for intermediate oxygen isotope values in surface dwelling foraminifera is pretty weak, only one specimen.

We agree that the evidence for intermediate values in oxygen isotopes is weak. Since focus in the literature has previously been on the carbon isotopes, we keep our discussion on this record. Incidentally, warming should be delayed relative to indicators for rapid carbon injection, potentially allowing for the preservation of intermediates in temperature proxies, though not in $\delta^{13}\text{C}$.

Line 67. This statement is not quite right. Grain size data suggest that sediments are winnowed.

We have clarified the text as

"Extensive sedimentary reworking has also been considered²⁵, given the evidence for bioturbation and possible winnowing in the core²⁶⁻²⁸. However, this is generally discounted as the sole explanation, largely on the basis of the preservation of abrupt step-like isotope shifts in the single species record^{17,20,26}"

Line 79. How was the modeled deep-water flow path determined?

We trace the deep-water flow path based on aging gradients reflected in modeled deep ocean $\delta^{13}\text{C}$, after Kirtland Turner and Ridgwell, 2013, (see References). We have clarified this in the manuscript.

Line 99 "We assign an among-individual standard deviation to the $\delta^{13}\text{C}$ values of each population (surface or thermocline) of 0.30‰ and 0.12‰, respectively." Do these values also come from Site 690 data?

These values are an approximation based on Site 690 data. We provide more detail in the supplement:

"There are a number of caveats that rightly make the probabilistic assessment of onset duration provided here a first-estimate, rather than the final word on the PETM onset duration. First, we had to define what it means for an individual to have a CIE-onset type isotopic value. Our modeled population variance in isotopes was less than that observed across all pre- (or peak-) event individuals in the thermocline or mixed layer. We used a relatively narrow variance, because we noted that within a sample, variance in the empirical record was relatively narrow and more like what we prescribe. The elevated variance across all pre-event (or peak-event) individuals appears to arise instead from slight variations in the mean with time –as is expected in an orbitally forced system."

Lines 122-129. I think this paragraph would be more powerful if there was a sentence stating that the pattern was similar to the observed data (Panel 1a).

Great! Done.

“Both the model simulations including realistic patterns of abundance change and the the simplified mixing model scenario produce isotopic distributions like those observed at ODP Site 690.”

Line 130-132. “Previous studies may have overlooked sediment mixing as an explanation for the Site 690 and 689 single-foraminifera records because of the non-intuitive effects that bioturbation can have on the fossil record.” “Overlooked” is not an appropriate word here. The Site 690 PETM is clearly bioturbated and several studies have considered the impact of bioturbation on the single specimen record (e.g., Zachos et al. (2007) and Bralower et al (2014). I would suggest, “underestimated the impact” as more appropriate.

This is a good suggestion. The text is now:

“Previous studies may have underestimated the effect of sediment mixing as an...”

Lines 377-78 Figure Caption for Figure 1. This diachronous step-change is readily simulated (b), assuming a greater change in the abundance of thermocline species (97.5%) than mixed-layer species (50%) (shown in panel (c)). It is hard to understand from this what these percentages mean.

Simplified by removing the %s from the caption.

Reviewer #2 (Remarks to the Author):

I very much would like to see a great paper emerge from this work, perhaps in Nature Communications. The authors have the talent and expertise to do this. However, I have mixed views on the current effort, and so I send a lengthy review of explanation for this perspective.

Sincerely,

Gerald (Jerry) Dickens

Prologue:

The PETM ca. 56 million years ago arguably represents our best past analog in which to understand rapid global warming and massive input of carbon to the ocean and atmosphere (Comment A). As might be expected for such an event, the causes and consequences remain the source of considerable interest and discussion across the broad Earth science community. Nature Communications should welcome good papers on the topic.

The current submission addresses broadly the timing of PETM carbon injection, and more narrowly, a puzzle regarding stable isotope analyses of individual foraminifera tests (shells) across the event. The PETM carbon input can be identified by a prominent negative carbon isotope excursion (CIE) in carbon-bearing phases. Individual foraminifera tests have been analyzed at several locations (including ODP Site 690 – the focus of the current study), but no tests have been identified so far with transitional $\delta^{13}\text{C}$ values (Zachos et al., Phil. Trans. Roy. Soc. A, 2007). The $\delta^{13}\text{C}$ of foraminifera are either pre-CIE or post-CIE (Comment B).

Some authors have interpreted the absence of transitional $\delta^{13}\text{C}$ values as signifying an extremely rapid input of ^{13}C -depleted carbon. However, as stressed by Zachos et al. (2007), there are at least three possibilities. Other than (i) “instantaneous” carbon input, there are (ii) dissolution of foraminifera tests because of corrosive waters on the seafloor (the original signal was removed), and (iii) diminished production of foraminifera tests because of environmental change in surface waters (an original signal was never generated).

We agree that Zachos et al., 2007 pointed out multiple possibilities to explain the Site 690 data. We have described these more clearly in the text.

The present manuscript sort of incorporates these thoughts into a numerical modeling perspective. The authors examine how a CIE should propagate through different reservoirs using a earth surface geochemical model

(albeit with key assumptions, Comment A, below), and then, assuming that foraminifera tests faithfully record water chemistry through the onset of the PETM, determine how the CIE would be recorded using a sedimentation model.

I stress “sort of” because the manuscript neither fully sets-up the problem, nor fully discusses alternative possibilities (multiple comments below). A root question to ask: can a more complete and effective effort be placed into Nature Communications?

Basic Issues:

(A) Location and mode of carbon input.

(A1) It is by no means clear that the massive input of carbon CAUSED the pronounced warming and environmental change across the PETM. Multiple explanations for the carbon injection exist. Some of these, for example release of carbon from peat or seafloor methane, imply that the input was a feedback to warming.

We recognize that some PETM explanations rely on carbon release as a threshold response to long-term warming. However, warming *during* the PETM is widely recognized as a response to greenhouse gas forcing. Our goal is to simulate abrupt carbon release that drove the CIE and the large warming that this carbon release caused, rather than simulating the cause of the initial carbon release itself.

(A2) It is by no means clear that the massive input of carbon ENTERED the atmosphere first. Some explanations, such as through dissociation of gas hydrate or North Atlantic sill intrusion, imply that much of carbon was added to the deep ocean.

We agree. However, injection of carbon to the deep ocean makes it even less likely that a delay in propagating the negative CIE from the surface to the deep ocean could be a real signal driven by changes in ocean circulation. To demonstrate this result, we have run a new suite of experiments varying the location of carbon injection. We now test carbon injection to the atmosphere, uniformly to the intermediate ocean around continental margins, and to the intermediate ocean in the North or South Atlantic. These results clearly indicate that atmospheric injection is the most likely to lead to a noticeable delay in propagating the CIE from the surface to the deep ocean at Site 690. When carbon is injected into the ocean, the result is that the thermocline and deep ocean see the excursion before the surface at modeled Site 690 — this is exactly the opposite of the Site 690 data. The only time we can get an excursion occurring first in the surface ocean and then thermocline was at Site U1403, corresponding to a location at the end of the deepwater flow path, when carbon is injected into the South Atlantic at intermediate depths. We have added discussion of these results to the text. (And also see results in Table S1 and Figs S1& S2).

The problem with the current effort is that plausible alternatives for carbon injection are not considered, and it is totally unclear how changes in basic assumptions would impact the modeling results and interpretations (Figures 2, 3).

We have added new experiments with alternative carbon injection scenarios and emphasize that the basic purpose of the Earth system modeling effort here is to demonstrate why a physical circulation mechanism to explain the Site 690 data is insufficient. This remains true regardless of whether carbon is injected into the atmosphere or ocean, but is even less likely if the carbon injection were submarine. The impacts of alternative *durations* of carbon injection were explored in detail in a previous manuscript (Kirtland Turner and Ridgwell, 2016, see References), and these experiments led us to realize that testing the most abrupt carbon release was the best way to push the system into generating large changes in ocean stratification. Longer durations of carbon release are simply further from offering a plausible circulation mechanism to explain the Site 690 data.

On these matters, I note two items. There are transitional values in the $\delta^{18}\text{O}$ of planktonic foraminifera at Site 690, a point emphasized by Kelly et al. (Geology, 2002). These authors emphasized that this suggests warming led carbon injection, an idea also promulgated in several other works. Dickens (Bull. Soc. Geol. France, 2000), using a relatively simple geochemical model, demonstrated that responses of parameters ($\delta^{18}\text{C}$, carbonate dissolution) in different reservoirs, depends strongly on the direction of carbon propagation (e.g., deep-water flow) and the location of carbon input.

Intermediate $\delta^{18}\text{O}$ values may indicate that warming led carbon injection, but it is also expected that warming should be delayed relative to the CIE (see Zeebe et al., 2016 in References). Such a delay could account for the appearance of transitional values in temperature even as the CIE has already reached a minimum. This is also clear from plots of the relative timing of the CIE and temperature change. We focus instead on the $\delta^{13}\text{C}$ data, which have presented the key conundrums.

(B) Background to the manuscript.

(B1) The CIE needs better definition and explanation. This is because, in various papers, the CIE represents the entire excursion (onset through recovery), or alternatively, as in this manuscript, the initial drop in $\delta^{13}\text{C}$ (onset of the PETM). However, even the latter aspect is complicated, because multiple records suggest a complex decrease in $\delta^{13}\text{C}$ over time. Indeed, the latter point is a “root problem” largely omitted in the present manuscript. While en vogue to discount bulk carbonate $\delta^{13}\text{C}$ records, the fact that similar records occur at widespread locations, including some sites lacking bioturbation (e.g., Mead Stream, Nicolo et al., Paleocean., 2010), raises a provocative issue. Why are bulk carbonate records, such as at Site 690 and several other locations, showing a gradual, multi-stepped onset, when the foraminifera records exhibit no such change? (See Comment B2).

We have tried to be clearer that what we are modeling and interested in is the initial drop in $\delta^{13}\text{C}$ rather than the full excursion and define CIE onset early in the text. The most obvious explanation for complex patterns of the CIE onset in bulk records is mixing. When populations of foraminifera are analyzed (and thus much more comparable to sampling to bulk carbonate records) there are intermediate values in the CIE onset preserved, just as is shown in bulk carbonate analyses. Of course, there is also the probability that carbon injection was not constant rate pulse, but rather a series of pulsed emissions of different rates. We showed the impact this would have on recorded CIEs in Kirtland Turner and Ridgwell, 2016 (see References). However, in this case, we sought to model the most extreme scenario that would be the most likely to lead to the observed Site 690 features. Also, as a comparison of the two halves of the 690 core show, the ‘steps’ in bulk isotopes only appear in one core half and are likely the result of sampling in-and-out of sediment burrows (e.g., see discussion in Bralower et al. 2014 in References). In other words, the steps mentioned are primary evidence for the importance of burrowing. As to why these appear in bulk records and not foraminifera, this has to do with the size of bulk toothpick samples (which fall within single burrows) relative to a foram-samples (which span burrows).

(B2) The “transitional foraminifera $\delta^{13}\text{C}$ puzzle” can be stressed better (see Zachos et al., 2007 and prologue above). More crucially, the alternative causes can and should be explored even further. What happens if the mixed-layer foraminifera examined for stable isotopes produced even less carbonate than modeled – in the end-member case – no carbonate during the onset of the PETM? Already, it is clear that multiple species of morozovellids and acarainids developed unusual morphologies near the start of the PETM.

We agree that carbonate reduction in the mixed layer dwellers is likely. This is the reason that we have reduced their population by 50% at the CIE onset in the sediment mixing model. We have emphasized in the text that the reduction of population size is applied to the sediment model and may therefore arise from changes in the living population size OR of the relative probability of dissolution in the water column, sediment column etc. as discussed extensively by Zachos et al. 2007 (see References). In the simplified model scenario (the only model scenario presented in the previous version of the manuscript), if the population of mixed layer dwellers declines by even more, then it would have been hard to fit the stratigraphic offset. However, as we describe (and have now tried to emphasize further) our results show the likelihood of sampling just the dominant sedimentary component. With a greater reduction in the population, then the individuals recording transitional values become an even smaller proportion of the total population. They are thus less likely to be sampled, and as a consequence, slightly longer ‘true’ onset durations may be interpreted as instantaneous.

The authors sort of note this point at the end of the Abstract (Lines 24-26), but then largely omit this crucial point from the rest of the manuscript. What if the mixed mixed-layer foraminifera examined for stable isotopes did not form carbonate during the onset of the PETM? This would solve many issues.

We have tried not to omit this point from the remainder of the manuscript. We agree: a key assumption is that there is not a total cessation of carbonate production by mixed layer dwellers at the event onset. However, our results show that there has to be a *different* drop in abundance between surface and thermocline to explain the lag in the CIE between these two populations. So, if mixed layer dwellers did not form carbonate at all, then this

may help explain a lack of intermediates, but it actually makes it more difficult to explain the stratigraphic delay, since we then need an even bigger impact on thermocline dwellers.

Other Problems:

(C) The modeling of carbon isotopes (Figure 2) needs better explanation in the text. The key point – and a good one – is that inversions in the mixed layer vs thermocline vs benthic $\delta^{13}\text{C}$ gradients can occur but cannot be maintained over time, despite stratification. However, I did not find this part crystal clear in the main text.

We agree that this is an important point and have tried to emphasize it in the revised text. However, it is important to understand that it is incorrect to say that inversions cannot be maintained, despite stratification. The point is that stratification itself cannot be maintained for long intervals.

(D) The issue of bioturbation is not fully discussed. One of the authors (Ridgwell) has emphasized, and I think correctly, that one of the keys to understanding deep-sea records across the PETM (with most sites coming from “intermediate” paleo-water depths) is a reduction in bioturbation. Yet, it is not clear to me, if and how such a reduction was incorporated into the sediment portion of the modeling.

Ridgwell has emphasized a reduction in bioturbation specifically to explain the intensity of dissolution, particularly at the Walvis Ridge sites, which show very large drops in wt% CaCO_3 across the PETM. The impacts of bioturbation on the $\delta^{13}\text{C}$ records were not considered (especially not single foram records since those studies did not use a sediment model capable of tracking individual particles). What matters to the sediment mixing modeling employed here (i.e., tracking the mixing up [and down] of pre-event and CIE individual particles) is the mixing intensity and sedimentation rate starting at the moment when the isotopic shift occurs and carrying on up into the CIE. Ridgwell and others have argued that mixing declined at around this moment (the CIE onset). And indeed, there is evidence for a change in mixing (a point we now discuss directly in text) in many PETM sections, including a very clear X-ray of the ODP Site 690 PETM core in the PhD thesis of Debbie Thomas. For the modeling we carry out, it is not the change across the pre-event/event that matters but rather that the mixing is going on with an effect at least reasonably approximated by our model. Without direct parameterization of point-events across the event (as from an ash fall), it is difficult to model a change in mixing intensity that is unparameterized.

It is worth noting that the decline in sedimentation rate would make the effect of even much reduced bioturbation depths and intensities look like that of higher intensity mixing under higher sedimentation rates. We have described this in detail in response to Reviewer 1 (please see above) and have now included multiple figures (Fig 3, S11 and S12) to make this point more intuitive. There is also a discussion of this point in the text.

Specific Comments:

(I could probably make more on a revised manuscript)

Line 11: This sentence needs rewriting. There is zero evidence to state that greenhouse gases caused global warming during the PETM (Comment A1).

We have to disagree here. Greenhouse gases clearly drove warming -- this is true regardless of evidence for preceding warming, but it is also most likely that preceding warming itself was driven by greenhouse gas release (albeit at a slower rate and possibly from a different source). We assume that this comment is not suggesting that release of greenhouse gases would not have driven warming.

Line 14: Several foraminifera from the PETM record at Site 690 do have intermediate values in $\delta^{18}\text{O}$ but not in $\delta^{13}\text{C}$, as noted above. Adding a word “... associated carbon isotope ...” (Line 13) would help to clarify.

Agreed. We have clarified that we are referring to the CIE.

Line 24-26: The notion that mixed layer foraminifera carbonate production needs to continue across the PETM is a really important concept. It needs to be highlighted later in the text.

Agreed. We have highlighted this later in the text.

Line 29: Should be “led to a prominent negative carbon isotope excursion”. This should be the very first point.

We have added the text ‘led to a prominent negative carbon isotope excursion’ as the first consequence of carbon addition.

Lines 29-30: Following from above, there is zero support to state that [a] “... rapid injection of ... carbon ... LED to a ~ 5°C global temperature rise, etc.” This is entirely wishful thinking with available data. “Associated with” is okay but “led” is not appropriate.

We have changed the wording to ‘associated with this is evidence for a ~5°C temperature rise...’

Lines 46-47: See above comment B1.

This comment refers to the definition of the CIE. We are not sure how it relates to the text in lines 46-47, but now we have added the following definition: “the interval between pre-PETM carbon isotope values and the recorded carbon isotope minimum.”

Line 50: I am not sure if this remains a correct statement. Maybe add “well-studied”.

Added.

Line 65-67: Needs rewriting. There are multiple reasons for why this is not diagenetic. More importantly, the observation has been made at several sites.

We have added more detailed explanation of why this is not diagenetic based on the analysis by Zachos et al., 2007 (see References). Crucially, the observation of the lack of intermediates has been made at other sites, but the observation of a large stratigraphic delay from surface to thermocline has been observed only at Sites 689 and 690.

Lines 75-79: This needs additional text. At the very least, it needs to be stated up front, that such modeling implicitly excludes certain possibilities for the PETM CIE excursion. (Ultimately, with a rapid carbon injection into the atmosphere from some “magical fossil-fuel source” that causes environmental change, they are invoking something absurd – like slow break-up of a giant methane-rich comet or a Mt. Pinatubo erupting every hour ...). Ideally, they fix and expand the framework to account for more reasonable possibilities.

We have modified the text to be explicit about the scenarios we are modeling and why. We have also added new experiments with carbon injection to the oceans (provided as supplementary figures S1 and S2). However, we emphasize that modeling a variety of rates of carbon emissions (as well as pulsed emissions) in GENIE was already comprehensively addressed in Kirtland Turner and Ridgwell, 2016 (see References). The purpose of the GENIE experiments presented in this paper is specifically to address the two conundrums of the 690 data. The controversy arises because these records have been interpreted as evidence for a very extreme PETM scenario. Therefore, we model an extreme scenario and argue, based on these results, that even such an extreme scenario (instantaneous PETM) cannot explain the records. However, this study is not meant to be a comprehensive evaluation of all PETM scenarios, especially as we recently published a similar exercise.

Lines 85-86: It might be useful to consider the modeling by Dickens (2000), as a supporting conclusion might be derived from this work.

We believe the reference is to carbon injection into different locations in the ocean in this paper. We have added a set of GENIE experiments with carbon injection to the intermediate oceans either uniformly around continental margins, or in the North or South Atlantic. The results do show differences in the carbon isotope records modeled for Site 690, but non-atmosphere injections actually move further away from being able to explain the ODP Site 690 data. We have added discussion/interpretation of these experiments to the text and cited Dickens, 2000.

Line 92: Needs rewording. It is gradual in time but not magnitude.

We have reworded this to say, “transforms into a more gradual decline to minimum $\delta^{13}\text{C}$ at depth.”

Line 97-98: I do not follow this. Why should the thermocline change occur in conjunction with the mixed layer? I would think that, with the model assumptions on input, changes would necessarily lag those in the mixed layer, albeit by not very much time.

Our GENIE modeling results indicate that there is virtually no delay between surface and thermocline at Site 690 (just a few years). This delay varies somewhat depending on where and how much carbon is injected as well as the simulated changes in ocean circulation. The sediment-mixing model has a depth step of 0.2cm (see Methods). In our simplified mixing model we assumed a sedimentation rate that matched the average rate across the interval (2.5cm/kyr). At 2.5 cm/kyr our sediment-mixing model steps in 80-year steps, so the offset in timing of onset between surface and thermocline does not exist for the sediment model as it is below the temporal resolution of the simulation.

Lines 103-104: A sentence or concept is missing here to arrive at the conclusion. Something like “But this is not observed.”

Changed.

Line 109-110: I do not fully follow the phrase “mixed-layer dwelling foraminifera (referred to as ‘robust’ acararinid variants in Ref . 29) and other acararinids.” There are two dominant genera of mixed layer foraminifera, morozovellids and acararinids, and both have “excursion taxa” across the PETM.

We are using the terminology of Kelly et al., 2002 (see References), who did the most comprehensive evaluations of faunal change at Site 690. Morozovellids first appear within the PETM at Site 690 (see Kelly et al., 2002 and Bralower et al., 2014 in References) following the appearance of so-called robust acararinids. However, because we have now added a second model scenario where we fit the empirical record of *Acaranina* and *Subbotina* abundance directly, we no longer justify the parameters for the ‘simplified scenario’ in this paragraph. Instead, we describe the difference between the two models and discuss the patterns of abundance change with regards to the second model scenario.

Lines 110-113: I do not follow the point of this sentence, as it includes the word “bulk”. This goes straight at comment B1. If the bulk carbon isotopes mean nothing, then how can be used to discuss stratigraphy?

As Kelly et al., 2002 (see References), pointed out, the issues raised by the single foram records did not alter the general biomagnetostratigraphic framework of the Site 690 PETM record. As a result, Kelly et al., 2002, used the bulk $\delta^{13}C$ stratigraphy to provide a “chemostratigraphic framework within which to evaluate faunal change.” In any case, in the revised manuscript this sentence has been deleted to make space for the discussion of the second model scenario, mentioned above.

Lines 122-123: The use of “sample” here is awkward. I assume a hypothetical sample.

We have changed the wording to ‘selected’ for clarity that this is a modeling exercise.

Lines 137-138: This sentence on sample sizes, while important, is not crystal clear.

This sentence was expanded for clarity to ‘Our small sampling size of just 4 individuals every 2 cm — while in fact slightly larger than empirical records — results in near-exclusive sampling of the dominant sedimentary component.’

Line 148: As above, there needs to be clarification of “samples”. In the current study, I think this refers to “model space”.

We have now changed the term to ‘modeled samples’ to avoid confusion.

Lines 155-157: This conclusion is problematic and absolutel needs rewriting. This is because, as noted above, it depends (I think) entirely on model assumptions pertaining to (i) carbon input, (ii) incorporation of bioturbation, and (iii) the fidelity of the foraminifera $\delta^{13}C$ record.

Lines 159-161: I would suggest that such efforts also point out where such modeling needs serious challenge.

We have made it clear that the results are model dependent, and we have explained the caveats involved in both the cGENIE and sediment mixing model results in much greater detail throughout this version of the manuscript. The sentence now reads:

“While the lack of intermediate values recorded at Site 690 does not require an instantaneous PETM onset, the simplified model of the Site 690 record does suggest that the onset likely occurred over < 5 kyr and not 10s of kyr (see Supplementary Discussion).

Lines 224-228: This is a fundamental problem with current cGENIE modeling. There is no means to add massive amounts of carbon rapidly without rubbing some bottle and invoking a magical source (perhaps why Andy called his model GENIE – ha ha), and no means to drive temperature without pushing from carbon. In particular, I do not think any one has suggested that pCO₂ increased to 10x let alone 25x pre-industrial values during the PETM. This should be stated as absurdly high. Basically, I can with these problems in modeling papers for a few more years, as long as they are acknowledged. Of course, it would be better if they were fixed.

It is true that cGENIE lacks a dynamic organic carbon reservoir that can generate large emissions of carbon; instead we must apply pulses of carbon of a specified mass and isotopic composition that may or may not be consistent with actual carbon reservoirs during the late Paleocene. Hopefully, ongoing model developments will allow us to test mechanisms behind carbon release from organic reservoirs, but such work is beyond the scope of the current study. However, it is incorrect that there is no way to drive temperature without pushing from carbon. We demonstrate this in our model by fixing radiative forcing. We have tried to make this methodology clear in the text – we are not actually changing pCO₂ to levels of x10 or x25 pre-industrial concentrations. We are just imposing the radiative forcing equivalent to those atmospheric pCO₂ levels. Basically, we just want to see the impact of extreme amounts of warming. This is a way to get around the imposed climate sensitivity in cGENIE. We can add a relatively small pulse of carbon (2275 Pg C of -60‰) but still evaluate the impact of much greater amounts of warming than the default cGENIE climate sensitivity would suggest in response to this carbon pulse. However, we do not suggest that climate sensitivity was this extreme. Our point is to demonstrate that even extreme warming still would not be able to generate the ocean stratification necessary to explain the ODP 690 data as reflective of ocean stratification.

Figures:

In the figures, the benthic CIE appears magnified relative to the mixed layer. This is interesting (and probably important), because it is not observed in available δ¹³C records.

We have run new experiments using a pulse of carbon rather than imposing a -4‰ CIE. This feature was an artifact of the previous experimental methodology. Now, the benthic CIE only appears larger at Site 690 for injection of carbon to the oceans at intermediate depth around continental margins.

For Figures S2 and S3: There is a time lag in temperature and δ¹³C propagation through ocean. It should be stated that the ordering is because of model assumptions. (See Dickens, 2000).

Our new experiments address the issue of how propagation timescales are influenced by the location of carbon injection.

Figure S4: I do not understand this figure conceptually. If all the carbon is added to the atmosphere to cause a 4 per mil excursion, then the excursion should be much, much less in the deep ocean once the carbon has propagated through the much larger ocean carbon reservoir (see for example, Zeebe et al. PNAS comment, 2014)..

We have changed the modeling methodology from the previous version of the manuscript. Before, we forced an instant and sustained -4 ‰ CIE in the atmosphere, which is not a PETM scenario per say, but was meant to

test the propagation of this anomaly from the sea surface. Now, to address more broadly the questions about how we've modeled the PETM, we use pulse emissions of a specified amount of carbon sufficient to generate a -4 ‰ CIE and apply the carbon in a single year either to the atmosphere or different ocean locations. These results should look more familiar to what one would expect for a PETM simulation. We have made all new supplementary figures to describe these new experiment results.

Supplementary Information:

Line 6: This should be "... isotope records constructed using single foraminifera at ODP Site 690."

Corrected.

REVIEWERS' COMMENTS:

Reviewer #1 (Remarks to the Author):

This is the second time I have reviewed this manuscript. In my previous review, I concluded that the manuscript represented a significant step forward in our understanding of the dynamics of the onset of the PETM. The GENIE and mixing model presented a viable solution to the conundrum raised by single foraminiferal isotope records at Site 690, arguably the best record of the event. At that time though I raised a number of concerns about assumptions in the model and the foraminiferal isotope data as well as the lack of clarity about the definition of the onset of the event. The authors have taken considerable time to adjust their experiments in response to my concerns as well as those of the other reviewer and I appreciate the careful letter that they have composed, the significant changes to the science and the manuscript, as well as some new figures that are very helpful in clarifying some of my previous misunderstandings. I believe that the manuscript now represents a major advance in our understanding of the PETM onset and merits publication with only a few minor corrections.

The only significant issue I have with the revised manuscript is the treatment of bulk carbonate isotope records. In lines 206-8, 374-5 (and Figure 4) the authors state that thermocline individuals contribute to bulk carbonate, but this is not true. The bulk record is composed 95% by nanoplankton that are in the < 10 micron size range and thus have the potential for greater mobility via bioturbation and winnowing than do the larger foraminifera. However, the relatively sharp onset of the bulk carbon isotope record contrasts with the smeared out single specimen records. I am guessing that this issue is entirely statistical, the bulk signal is comprised of literally millions of nanofossils as opposed to a handful of single specimens, and so the mixing does not show up in the stratigraphy of bulk carbonate. I am not proposing that the authors try to model the bulk record because that is a very large undertaking, but I do recommend that they correct the text so that the difference between foraminiferal and nanofossil records are noted.

A more minor point, I still believe a 97.5% reduction in *Subbotina* abundance (line 275) is unlikely without some significant dissolution which is not included in the model. Again, this is something for future research, not for this version.

Minor points

Line 47-49, the authors need to explain why single specimen records can be used in this fashion. This might not be apparent to someone who has not thought about this issue.

Line 133 "modeled" instead of "model"

Line 147 "site" not "Site"

Line 169 "ODP Site 690 data"

Reviewer #2 (Remarks to the Author):

I have read through all the commentary, and I am okay with the current submission and think should be published. I do not agree with all ideas, but the revised MS discusses things in proper context and makes the community think.

Jerry

Response to Reviewers

The only significant issue I have with the revised manuscript is the treatment of bulk carbonate isotope records. In lines 206-8, 374-5 (and Figure 4) the authors state that thermocline individuals contribute to bulk carbonate, but this is not true. The bulk record is composed 95% by nanoplankton that are in the < 10 micron size range and thus have the potential for greater mobility via bioturbation and winnowing than do the larger foraminifera. However, the relatively sharp onset of the bulk carbon isotope record contrasts with the smeared out single specimen records. I am guessing that this issue is entirely statistical, the bulk signal is comprised of literally millions of nannofossils as opposed to a handful of single specimens, and so the mixing does not show up in the stratigraphy of bulk carbonate. I am not proposing that the authors try to model the bulk record because that is a very large undertaking, but I do recommend that they correct the text so that the difference between foraminiferal and nannofossil records are noted.

We apologize, this again was a problem of wording rather than modeling. We did not mean to imply that there was any relationship between the thermocline and bulk carbonate individuals, but rather that a similar type of change in population abundance could produce a realistic looking abundance trajectory. To clarify, the first sentence now reads:

“We can similarly reproduce the gradual decline observed in bulk carbonate isotopic values (Fig. 4a) with an abrupt reduction of nanoplankton populations synchronous with the PETM onset followed by bioturbation.”

and latter reads:

“Modeled bulk carbonate (Fig. 4b) was modeled as the mean of a population experiencing a isotopic shift from 1.6 to -0.4‰ and a total abundance change to levels less than 1% of pre-event levels for 10cm following the PETM onset and 2.5% of pre-event levels thereafter.”

and in the Figure 4 caption:

“ assuming a greater change in the abundance of thermocline species and in the nanoplankton taxa comprising the majority of the bulk carbonate than in mixed-layer species.”

We recognize that the edits to the second sentence make an already complicated sentence longer, but hopefully also clearer, as it contains the necessary details of the modeling without appearing to tie the changes in nanoplankton to that in the Subbotinids, which was never our intention.

Getting to the specifics of your question (modeling bulk carbonate being a large undertaking), we actually already track 10,000 particles per stratigraphic level, so the modeling is fitting for the problem of tracing the nanoplankton pattern as well as foraminifera. There is no fundamental difference in the model needed to track these two different groups, so we have left the text as revised above and hope that it helps avoid similar confusion in our readers.

A more minor point, I still believe a 97.5% reduction in Subbotina abundance (line 275) is unlikely without some significant dissolution which is not included in the model. Again, this is something for future research, not for this version.

Yes, it would be interesting in a different sort of exercise to see this issue of dissolution versus population change considered! Within our modeling framework, the cause of the population decline isn't testable, so this is certainly an area for future investigations.

Minor points

Line 47-49, the authors need to explain why single specimen records can be used in this fashion. This might not be apparent to someone who has not thought about this issue.

We added the phrase “because of the short foraminiferal lifespan” to clarify why individual foraminifera tests should enable detection of rapid environmental change.

Line 133 “modeled” instead of “model”

Corrected

Line 147 “site” not “Site”

Corrected

Line 169 “ODP Site 690 data”

Corrected